# A tissue-bioengineering strategy for modeling rare human kidney diseases in vivo

J. O. R. Hernandez[1,8], X. Wang[1,8], M. Vazquez-Segoviano[1], M. Lopez-Marfil [1], M. F. Sobral-Reyes[1], A. Moran-Horowich[1], M. Sundberg [2,3], D. O. Lopez-Cantu[1], C. K. Probst[4,5], G. U. Ruiz-Esparza[1], K. Giannikou[3,4,5], R. Abdi[6], E. P. Henske[3,4,5], D. J. Kwiatkowski [3,4,5], M. Sahin [2,3] & D. R. Lemos [1,3,7 ✉]

The lack of animal models for some human diseases precludes our understanding of disease mechanisms and our ability to test prospective therapies in vivo. Generation of kidney organoids from Tuberous Sclerosis Complex (TSC) patient-derived-hiPSCs allows us to recapitulate a rare kidney tumor called angiomyolipoma (AML). Organoids derived from $TSC2^{-/-}$ hiPSCs but not from isogenic $TSC2^{+/-}$ or $TSC2^{+/+}$ hiPSCs share a common transcriptional signature and a myomelanocytic cell phenotype with kidney AMLs, and develop epithelial cysts, replicating two major TSC-associated kidney lesions driven by genetic mechanisms that cannot be consistently recapitulated with transgenic mice. Transplantation of multiple $TSC2^{-/-}$ renal organoids into the kidneys of immunodeficient rats allows us to model AML in vivo for the study of tumor mechanisms, and to test the efficacy of rapamycin-loaded nanoparticles as an approach to rapidly ablate AMLs. Collectively, our experimental approaches represent an innovative and scalable tissue-bioengineering strategy for modeling rare kidney disease in vivo.

[1] Renal Division, Brigham and Women's Hospital, Boston, MA 02115, USA. [2] Rosamund Zander Stone Translational Neuroscience Center, Department of Neurology, Boston Children's Hospital, Boston, MA 02115, USA. [3] Harvard Medical School, Boston, MA 02115, USA. [4] Cancer Genetics Lab, Division of Pulmonary and Critical Care Medicine, Brigham and Women's Hospital, Boston, MA 02115, USA. [5] Center for LAM Research and Clinical Care, Division of Pulmonary and Critical Care Medicine, Brigham and Women's Hospital, Boston, MA 02115, USA. [6] Transplantation Research Center, Renal Division, Brigham and Women's Hospital, Harvard Medical School, Boston, MA, USA. [7] Harvard Stem Cell Institute, Cambridge, MA 02138, USA. [8]These authors contributed equally: J.O.R. Hernandez, X. Wang. ✉email: dlemos@bwh.harvard.edu

Certain human diseases are driven by genetic and cellular mechanisms that cannot be recapitulated using transgenic mouse models. The lack of an in vivo model represents a limitation for the elucidation of disease mechanisms, and for testing prospective therapies on a preclinical level. One such disease is renal angiomyolipoma (AML), a tumor found in 80% of patients with Tuberous Sclerosis Complex (TSC). AML growth can cause kidney failure and lead to premature death due to the formation of vascular aneurysms prone to spontaneous bleeding[1,2]. AMLs have been classified as perivascular epithelioid cell tumors, or PEComas, a term that designates mesenchymal neoplasms composed of anatomically and immunohistochemically distinctive myoid cells expressing smooth muscle and melanogenic genes[3–6].

Rapamaycin analogs (rapalogs) are the main treatment for AML. Rapalogs reduce tumor size by inhibiting the dysregulated activity of mTORC1 that results from biallelic inactivation of *TSC1* or *TSC2* and loss of activity of the encoded proteins, namely TSC1 (hamartin) or TSC2 (tuberin), which act as repressors of the metabolic activator kinase mTOR[7]. In their current form, however, rapalog therapies rarely abrogate AML completely, making life-long treatment necessary. Relapse often occurs after treatment is interrupted due to adverse effects that may include proteinuria, oral ulcers, dyslipidemia and pneumonitis[8–11]. The lack of animal models has been an obstacle for the development of therapies and for the improvement of rapalog-based treatments.

The etiological mechanisms driving AML are difficult to recapitulate with transgenic animal models. AMLs are initiated by a Knudson's two-hit mechanism of tumorigenesis following a second copy neutral loss-of-heterozygosity (LOH) mutation in the *TSC1* or, more commonly, in the *TSC2* locus, in addition to a pre-existing germline inactivating mutation[12,13]. The second hit results in biallelic inactivation of either *TSC1* or *TSC2*, and constitutive mTOR activation, driving anabolic cell metabolism and aberrant tissue growth[14]. Efforts to recapitulate these mechanisms in vivo have been challenging due to the fact that biallelic inactivation of *TSC1* or *TSC2* during development causes embryonic lethality. In addition, the developmental mechanisms giving raise to AML lesions has remained elusive[4,15], contributing to the lack of success in previous attempts to ablate *TSC1* or *TSC2* by means of tissue-specific Cre-mediated recombination[16,17]. Limited success was achieved using an in vivo approach that involved low-dose doxycycline-induced DNA recombination events to ablate *Tsc1* stochastically in mice carrying a ubiquitously expressed Cre transgenic allele[18]. Using this strategy, small kidney lesions with characteristics of AMLs could be detected, but the resulting number of mice carrying lesions was small, reducing the suitability of this model for either mechanistic or drug-testing studies[18].

In this work, we present a tissue-bioengineering approach for modeling TSC-associated human kidney diseases in vivo, using transplanted hiPSC-derived renal AML organoids. We show that nephric differentiation of patient-derived, and genetically-edited *TSC2⁻/⁻* hiPSCs but not of isogenic *TSC2⁺/⁻* hiPSCs[19], resulted in formation of two-dimensional (2D) and three-dimensional (3D) kidney tissues recapitulating TSC-associated AML and cystic disease. Transplantation of multiple 3D *TSC2⁻/⁻* organoids into the kidneys of immunodeficient rats results in fully vascularized human AML xenografts that permitted us to identify tumor resistance mechanisms, and to test the efficacy of an in situ approach for rapid tumor ablation. The methodology presented here is broadly applicable for the study of other rare kidney diseases for which no in vivo experimental model currently exists.

## Results

**Nephric differentiation of *TSC2⁻/⁻* hiPSCs results in renal vesicle cultures containing myoid cells with a melanogenic phenotype.** To understand how lack of TSC2 affects the early stages of nephric differentiation, we used an in vitro protocol previously developed for the directed differentiation of hiPSCs into kidney tissues[20]. We used a set of three isogenic hiPSC lines that included a line derived from an individual carrying a heterozygous 9-bp deletion in the *TSC2* locus (*TSC2⁺/⁻*), a second isogenic *TSC2⁻/⁻* hiPSC line that was generated by introducing a TALEN-engineered second deletion in the wild type (WT) *TSC2* allele of the patient-derived hiPSC line[19], and lastly a *TSC2⁺/⁺* hiPSC line in which the original *TSC2* deletion present in the patient-derived hiPSC line was corrected using CRISPR-Cas9 (Fig. 1a)[19]. The *TSC2⁻/⁻* hiPSC line displayed significantly increased mTOR activity compared to the *TSC2⁺/⁻* and *TSC2⁺/⁺* hiPSC lines, as indicated by the levels of phosphorylation of the ribosomal protein S6 (Phospho-S6) detected by Western Blot (Fig. 1b). No differences in phospho-S6 levels were observed between the *TSC2⁺/⁻* and *TSC2⁺/⁺* hiPSC lines, a finding that was consistent with the idea that one functional *TSC2* allele is sufficient to preserve mTOR activity under these conditions (Fig. 1b).

To generate renal tissues from the three hiPSC lines in 2D conditions, we used a previously established protocol that involves an initial step of metanephric mesenchyme induction by initial incubation with the WNT activator CHIR99201 (CHIR) at 8 μM for 4 days, followed by incubation with 20 ng/ml Activin A for three days (Fig. 1c) and subsequent incubation with 25 ng/ml fibroblast growth factor 9 (FGF9) for two days[20]. On Day 9, the 2D cultures received a 1-h pulse of 3μM CHIR. The 8-μM concentration of CHIR during the initial induction phase was chosen based on comparable mRNA expression profiles observed for the nephric lineage homeobox genes *SIX2* and *SALL1* in the three hiPSC genotypes between Day 0 and Day 9, which were determined by quantitative polymerase chain reaction (qPCR) (Fig. 1d). Both SIX2 and SALL1 were also detected by immunofluorescence in cultures of the three hiPSC lines, confirming induction of these nephric lineage markers (Supplementary Fig. 1a). Microscopic field quantification indicated that ~82, 85, and 91% of *TSC2⁺/⁻*, *TSC2⁻/⁻*, and *TSC2⁺/⁺* cells, respectively, were SIX2⁺; and ~80, 83, and 87% of *TSC2⁺/⁻*, *TSC2⁻/⁻*, and *TSC2⁺/⁺* cells, respectively, were SALL1⁺, indicating efficient induction of all three cell lines (Supplementary Fig. 1a). After confirming induction of metanephric mesenchyme from the three hiPSC lines, we assessed their ability to subsequently form the renal vesicles that give rise to nephrons. To that end, we continued incubating the metanephric mesenchyme cultures with FGF9 until Day 14. During this stage, the cultures were exposed to a 24-h pulse of 1μM retinoic acid on Day 13 of differentiation (Fig. 1c). This modification to the original protocol was adopted from a recent protocol for the differentiation of podocytes[21] and was introduced based on our observation of glomeruli of an increased size present in Day 21 cell cultures (Supplementary Fig. 1b, c). In order to assess the formation of renal vesicles on Day 14 we used a combination of immunostaining for the homeobox transcription factor PAX8 and qPCRs to detect the expression of the nephric homeobox genes *PAX8*, *LHX1*, *HNF1*, and *WT1*. We confirmed the formation of cell aggregates with the morphology of renal vesicles containing PAX8⁺ renal progenitor cells (Fig. 1f). In addition, we detected similar expression levels for renal vesicle markers in *TSC2⁻/⁻*, *TSC2⁺/⁻*, and *TSC2⁺/⁺* cell cultures, and found that all were upregulated compared to undifferentiated hiPSC cultures (Fig. 1g).

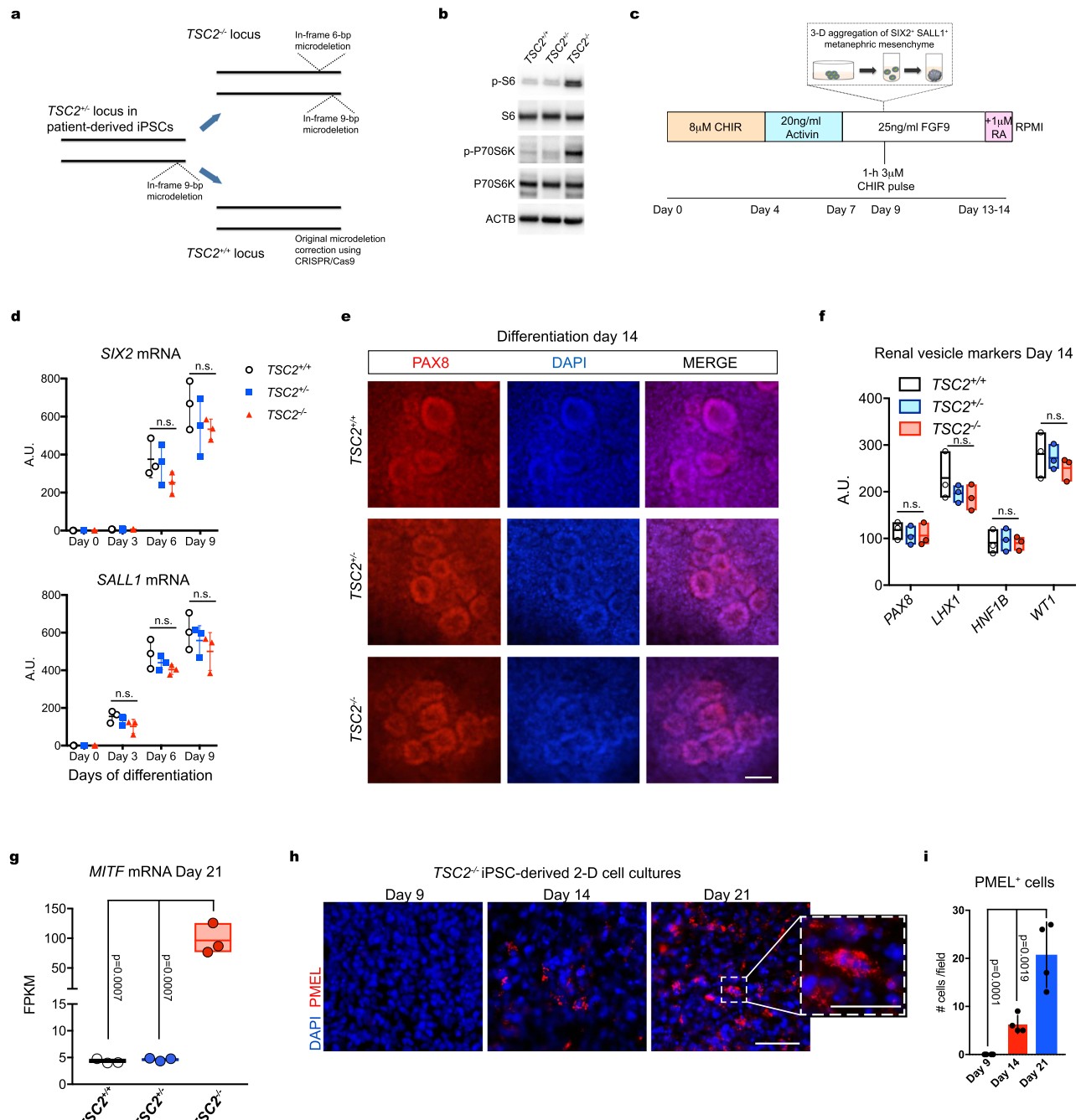

Given that loss of *TSC1* or *TSC2* is a common occurrence in kidney AMLs from TSC patients[13], we next performed global gene expression analysis comparison by means of whole transcriptome RNA sequencing (RNA-seq) of *TSC2+/−*, *TSC2+/+*, and *TSC2−/−* iPSC-derived organoids generated by differentiation of Day-9 metanephric mesenchymal cell aggregates in three-dimensional culture conditions[22]. Our analysis revealed that the AML master driver oncogene *MITF*[23] was significantly upregulated in *TSC2−/−* iPSC-derived Day-21 organoids, compared to *TSC2+/−* and *TSC2+/+* iPSC-derived organoids (Fig. 1g). Using the mouse monoclonal antibody HMB45 which is routinely used in clinical histopathology for diagnosing AML[5,24], we detected cells expressing the melanogenic protein premelanosome 17 (PMEL) in differentiating 2D cultures of *TSC2−/−* iPSC-derived cells but not in *TSC2+/−*, *TSC2+/+* iPSC-derived cell cultures (Fig. 1h). We first detected *TSC2−/−* PMEL-expressing cells on differentiation Day

14 and determined that their number increased by Day 21 (Fig. 1i). Of note, *TSC2−/−* PMEL+ cells had a plump myoid morphology (Fig. 1i), distinct from the typical dendritic morphology of epidermal melanocytes in culture[25,26].

Taken together, these results indicated that while loss of *TSC2* is not an impediment for the nephric specification of iPSCs into metanephric mesenchyme and formation of renal vesicles, the differentiation results in the generation of cells resembling the myoid melanogenic phenotype observed in kidney AML cells[3,24].

**TSC2−/− hiPSCs generate 3D renal organoids with characteristics of kidney AMLs in vitro.** Our results indicating the specification of cells with myoid/ melanocytic characteristics throughout our protocol of nephric differentiation prompted us to investigate kidney AML features in *TSC2−/−* iPSC-derived

**Fig. 1 Nephric differentiation of TSC2$^{-/-}$ hiPSCs results in generation of myoid melanocyte-like cells. a** Schematic representation of the gene-editing strategies used to either introduce a second inactivating mutation in the wild type allele of patient-derived *TSC2$^{+/-}$* iPSCs, or to correct the original deletion mutation. **b** Immunoblots showing phosphorylation of S6 and p-70S6K in *TSC2$^{+/+}$*, *TSC2$^{-/-}$*, and *TSC2$^{+/-}$* hiPSCs, indicating mTORC1 activation in the *TSC2$^{-/-}$* cells. Similar results were obtained in two independent experiments. **c** Schematic representation for the hiPSC nephric differentiation protocol used to generate 2D renal tissues and 3D renal organoids. CHIR: CHIR99021; FGF9: fibroblast growth factor 9. **d** Quantitative real-time-polymerase chain reaction (RT-PCR) analysis of *SIX2* and *SALL1* mRNAs during the phase of metanephric mesenchyme induction. Curve points represent mean ± SD in arbitrary units (A.U. normalized to β-actin), $n = 3$ independent technical replicates. Nonsignificant mean differences determined by two-way ANOVA analysis using Tukey test for multiple comparisons are indicated as n.s. **e** Representative immunofluorescence images showing PAX8 expression in 2D *TSC2$^{+/+}$*, *TSC2$^{-/-}$*, and *TSC2$^{+/-}$* cell cultures on Day 14 of differentiation. Scale bar, 50 μm. **f** Quantitative RT-PCR analysis of *PAX8, LHX1, HNF1B* and *WT1* mRNAs in 2D *TSC2$^{+/+}$*, *TSC2$^{-/-}$* and *TSC2$^{+/-}$* cell cultures on differentiation Day 14, showing similar expression independent of genotype. Floating bars graph represent mean ± SD, $n = 3$ independent experiments. Nonsignificant mean differences determined by two-way ANOVA analysis using Tukey test for multiple comparisons are indicated as n.s. **g** Expression of the AML transcription factor *MITF* as determined by RNAseq analysis of *TSC2$^{+/+}$*, *TSC2$^{+/-}$*, and *TSC2$^{-/-}$* iPSC-derived renal organoids on differentiation Day 21. Floating bar graphs represent mean ± SD, $n = 3$ samples for each genotype, five organoids *per* sample. Gene expression is shown in fragments per kilo base per million mapped reads (FPKM) values. *P* values for individual comparisons done using Tukey test in one-way ANOVA analysis are indicated. **h** Representative immunofluorescence images of 2D *TSC2$^{-/-}$* iPSC-derived cell cultures showing expression of PMEL on differentiation Days 9, 14, and 21. Scale bars, 25 μm and 12.5 μm for high magnification image. **i** Bar graphs showing the quantification of PMEL$^+$ cells in immunofluorescence images taken at Days 9, 14, and 21 of differentiation in 2D conditions. Values in the bar graph represent mean ± SD, $n = 4$ experiments. *P* values for individual comparisons done using Tukey test in one-way ANOVA analysis are indicated.

organoids. Our immunofluorescence analysis did not detect PMEL$^+$ cells in *TSC2$^{+/-}$* and *TSC2$^{+/+}$* kidney organoids (Fig. 2d), a finding that was also consistent with a previous study using the Morizane protocol for nephric differentiation of hPSCs[27]. By contrast, and consistent with our analysis of Day-21 2D *TSC2$^{-/-}$* cell cultures, we identified a large population of PMEL$^+$ cells in *TSC2$^{-/-}$* organoids, covering an area that was approximately 6.2% of the organoid sections analyzed (Fig. 2a). PMEL$^+$ cells were present in the interstitial space, but were not peritubular, a characteristic that distinguished them from renal fibroblasts observed in both human kidneys and hPSC-derived organoids. Similar to what we observed in our 2D cultures, 3D cultured PMEL$^+$ cells had a myoid spindle-like morphology with a granular staining pattern that resembled the characteristic phenotype of human kidney AML cells (Fig. 2a, Supplementary Fig. 2a)[3]. Co-expression of PMEL and glycoprotein NMB (GPNMB), another AML cell marker[28], further confirmed the melanogenic phenotype of *TSC2$^{-/-}$* organoid cells (Fig. 2b). Next, we investigated the expression and distribution of cells expressing actin alpha 2, smooth muscle (ACTA2, commonly known as alpha-smooth muscle actin), whose aberrantly high expression is a hallmark of AML[3], and which is expressed at low levels in non-injured kidneys and WT organoids[29]. We observed that ACTA2 was highly expressed in *TSC2$^{-/-}$* organoids, with ACTA2$^+$ cells covering an area that represented 21.7% of the sections analyzed, compared to 3.28 and 3.7% of the areas in *TSC2$^{+/+}$* and *TSC2$^{+/-}$* organoid sections, respectively (Fig. 2c). Similar to what we observed for PMEL$^+$ cells, ACTA2$^+$ cells were sparsely distributed throughout the stroma of *TSC2$^{-/-}$* organoids (Fig. 2c). Consistent with what has been previously reported, the ACTA2$^+$ cells observed in *TSC2$^{+/-}$* and *TSC2$^{+/+}$* organoids were mostly confined to regions surrounding nephron epithelial structures (Supplementary Fig. 2b). Morphologically, ACTA2$^+$ cells found in *TSC2$^{-/-}$* organoids had a rounder and myoid morphology that best matched the well-characterized morphology of ACTA2$^+$ cells observed in kidney AMLs (Fig. 2c, Supplementary Fig. 2a)[15]. By contrast, ACTA2$^+$ cells in *TSC2$^{+/-}$* and *TSC2$^{+/+}$* organoids presented an elongated spindle-like cell body with extended body processes resembling the morphology of kidney fibroblasts (Supplementary Fig. 2b)[29]. Given that co-expression of melanogenic and mesenchymal cell genes is unique to AML cells, we next investigated whether *TSC2$^{-/-}$* organoid cells had a myomelanocytic phenotype. Using immuno-fluorescence we were able to establish that ~94% of GPNMB-expressing cells were ACTA2$^+$ (Fig. 2d). Cells co-expressing both markers were also detected in AML samples, but not in activated

fibroblasts/myofibroblasts from injured *TSC2$^{+/+}$* organoids. In order to validate the myomelanocytic cell phenotype using a different technical approach, we analyzed the distribution of PMEL$^+$ and ACTA2$^+$ cell populations in *TSC2$^{-/-}$* organoids by means of fluorescence-activated cell sorting (FACs). Our analysis identified a population of ACTA2$^+$ cells that represented approximately 31.2% of the total cells recorded in *TSC2$^{-/-}$* organoid samples (Fig. 2e, Supplementary Fig. 3a). The same cell population constituted 3.5 and 4.1% of the cells recorded in *TSC2$^{+/-}$* and *TSC2$^{+/+}$* organoid samples, respectively (Fig. 2e). Within the ACTA2$^+$ cell population, approximately 24.3% of the cells also expressed PMEL (Fig. 2e), whereas no PMEL$^+$ ACTA2$^-$ cells were detected, a finding that was consistent with our immunofluorescence results for GPNMB and ACTA2 (Fig. 2d). Immunofluorescence tracking experiments in 2D cultures further indicated that *TSC2$^{-/-}$* myomelanocytic cells appeared by Day 14 of iPSC differentiation coinciding with the time of formation of renal vesicles (Supplementary Fig. 2c). We next compared the expression of myomelanocytic cell markers in prospectively purified live GPNMB$^+$ cells *versus* GPNMB$^-$ cell fractions of *TSC2$^{-/-}$* organoids (Figure f, Supplementary Fig. 3b). We detected significant expression of AML markers *MLANA*, *GPNMB*, and *CTSK* in the GPNMB$^+$ fraction but not in the GPNMB$^-$ fraction (Figure f). Expression of ACTA2 was detected in both cell populations, albeit more abundantly in the GPNMB$^-$ cell fraction (Figure f), a result that was consistent with our FACs analysis results of PMEL and ACTA cell fractions. Taken together, our histology, FACS and gene expression data identified a population of *TSC2$^{-/-}$* organoid cells that recapitulates the unique phenotype and gene expression profile observed in kidney AML myomelanocytic cells[15]. Additionally, we confirmed the abnormal expression of GPNMB and ACTA2 in *TSC2$^{-/-}$* organoids by means of Western Blot performed on protein extracts from *TSC2$^{+/-}$* and *TSC2$^{+/+}$* and *TSC2$^{-/-}$* organoids (Fig. 2g).

Collectively these in vitro data indicate that renal organoids derived from *TSC2$^{-/-}$* hiPSCs, but not from *TSC2$^{+/-}$* hiPSCs, possessed several key characteristics of renal AML tumors, including the presence of myomelanocytic cells with a typical AML morphology. These results also provide critical experimental support to the hypothesis that the signals driving AML are present during renal development.

**Gene expression analysis of *TSC2$^{-/-}$* renal organoids validates an AML identity.** To investigate how loss of *TSC2* affects global gene expression in hiPSC-derived renal organoids, we performed pairwise DESeq2 differential gene expression analyses of our whole

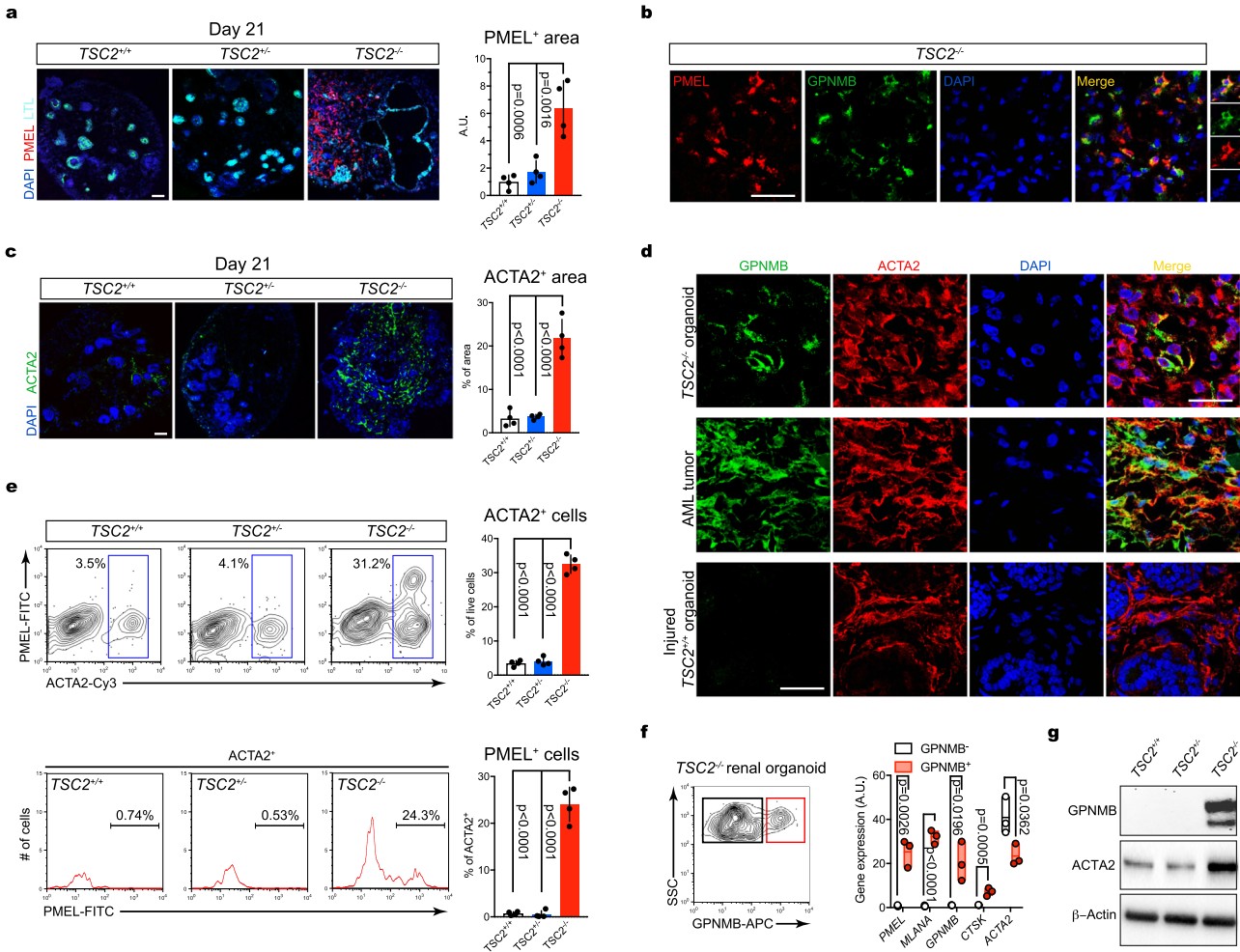

**Fig. 2 TSC2$^{-/-}$ renal organoids recapitulate human AML tumor phenotype. a–c** Confocal optical sections showing LTL with PMEL (**a**), PMEL and GPNMB (**b**), and ACTA2 (**c**) in Day 21 3D renal organoids derived from TSC2$^{+/+}$, TSC2$^{-/-}$ and TSC2$^{+/-}$ hiPSCs. Bar graphs in (**a**) and (**c**) represent mean ± SD, $n = 4$ experiments. P values for individual comparisons done using Tukey test in one-way ANOVA analysis are indicated. **d** Representative confocal immunofluorescence sections showing expression of GPNMB and ACTA2 in Day-21 TSC2$^{-/-}$ renal organoids, kidney AML tumor and fibrotic Day-28 TSC$^{+/+}$ renal organoids injured by incubation with interleukin 1β for 96 h. **e** Representative FACs plots and quantification showing the percentage of ACTA2$^+$ cells and PMEL$^+$ cells in Day-21 TSC2$^{+/+}$, TSC2$^{-/-}$ and TSC2$^{+/-}$ renal organoids. Bar graphs represent mean ± SD, $n = 4$ experiments. P values for individual comparisons done using Tukey test in one-way ANOVA analysis are indicated. **f** Representative cell sorting plot showing the GPNMB$^+$ (red square) and GPNMB$^-$ (black square) cell fractions in TSC2$^{-/-}$ renal organoids and gene expression levels for PMEL, MLANA, GPNMB, CTSK, and ACTA2, determined by Quantitative RT-PCR analysis. Floating bars graph represent mean ± SD, $n = 3$ independent experiments, containing three organoids each. P values for individual comparisons done using two-tailed Student's t test are indicated. **g** Representative immunoblot showing expression of GPNMB and ACTA2 in renal organoids from the three genotypes. Two independent experiments were performed with similar results. Scale bars, 50 μm.

transcriptome RNA-seq datasets, comparing TSC2$^{-/-}$ organoids to TSC2$^{+/+}$ and TSC2$^{+/-}$ organoids, using Qlucore software. Several hundred genes were upregulated or downregulated at a false discovery rate (FDR)/ q value < 0.05, p value < 0.002, |log$_2$ fold | > 2 and |log$_2$ fold change | <-2 (Supplementary Table 1). Among the genes upregulated in the TSC2$^{-/-}$ organoids were: MLANA, PMEL, GPNMB, CTSK, and ACTA2, with median expression fold change 121x-, 88x-, 34x-, 14x-, 9.5x, respectively, all of which are known to be highly expressed in the renal angiomyolipoma (Fig. 3a). In contrast, DEseq2 analysis of TSC2$^{+/+}$ vs. TSC2$^{+/-}$ renal organoids did not reveal remarkable differences. Multiple housekeeping genes, including GPDH, GUSB, RPL19, showed no significant difference or minimal changes in expression across the three kidney organoid genotypes providing further confirmation that the changes observed were specific to the genes identified (Supplementary Fig. 4a).

Principal Component Analysis (PCA) of RNA-Seq data for all three genotypes also demonstrated that the TSC2$^{-/-}$ organoids were transcriptionally different from TSC2$^{+/+}$ and TSC2$^{+/-}$

kidney organoids, which clustered together (Fig. 3b). Hierarchical clustering using Spearman rank correlation for the 3000 most variable genes also demonstrated that the TSC2$^{-/-}$ organoids were distinct from the other two genotypes, with the latter appearing similar (Fig. 3c). Gene set enrichment analysis (GSEA) was performed comparing TSC2$^{-/-}$ renal organoid RNA data to that of TSC2$^{+/+}$ renal organoids, and showed enrichment in multiple gene sets at FDR/q < 0.25, with similar results in comparison of TSC2$^{-/-}$ and TSC2$^{+/-}$ renal organoids. Hallmark gene sets enriched in expression in TSC2$^{-/-}$ renal organoids in each comparison included IL6-JAK-STAT3 signaling, adipogenesis, angiogenesis, fatty acid metabolism, KRAS signaling, and estrogen response (Fig. 3d, Supplementary Fig. 4a, b).

We then investigated the similarity between the genes identified in the comparisons between TSC2$^{-/-}$ vs. TSC2$^{+/+}$ organoids against differential genes identified in the comparison between kidney AML vs. normal kidney (Fig. 3e). The 187 common differentially expressed genes included the AML

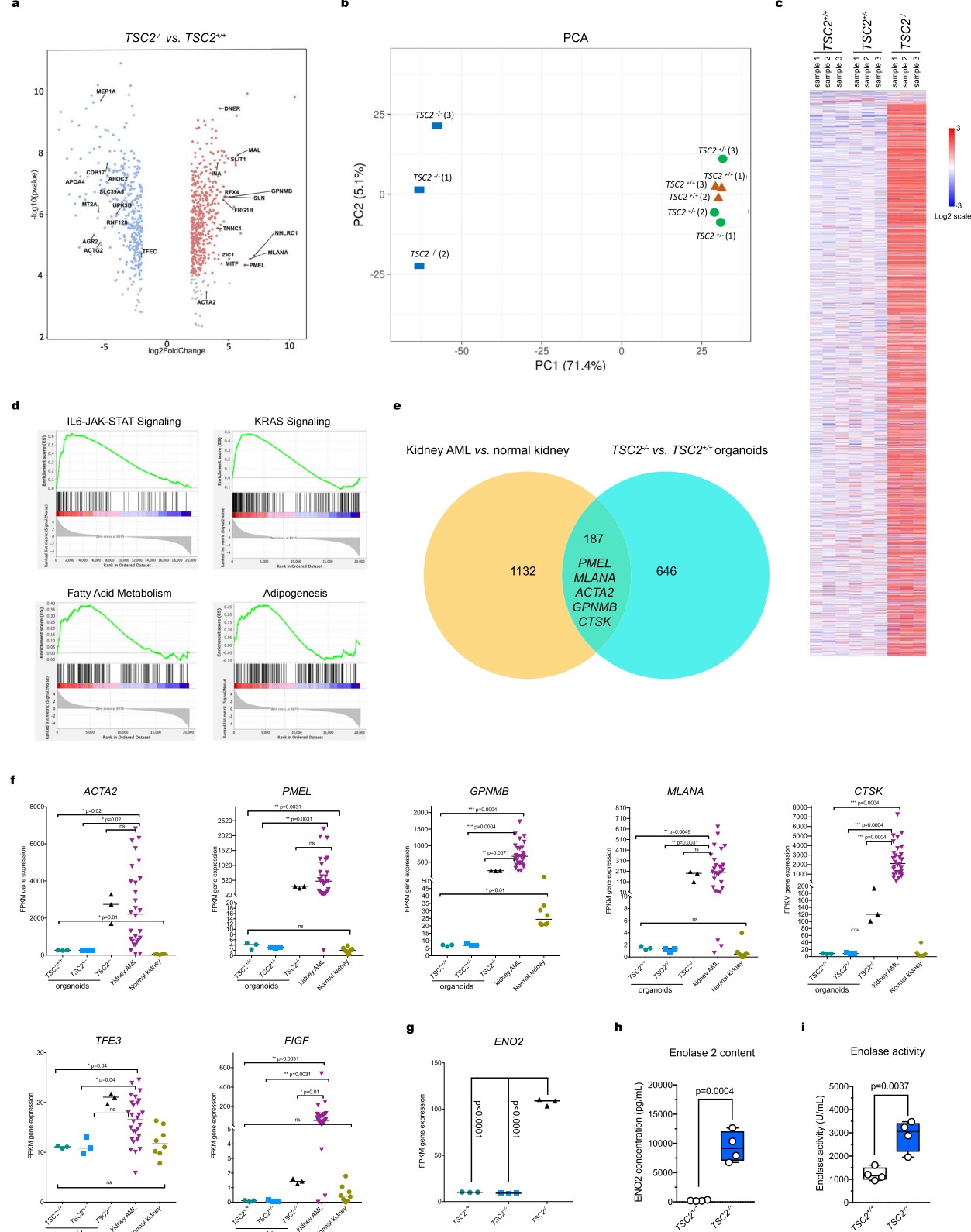

hallmark genes *MLANA, PMEL, GPNMB, MITF, CTSK,* and *ACTA2,* expanding the similarities observed between $TSC2^{-/-}$ organoids and kidney AML to other signature markers (Fig. 3e). Direct comparison of mRNA expression levels for those markers plus the AML transcription factors *TFE3* and *FIGF* (Fig. 3f) confirmed a similar expression profile for $TSC2^{-/-}$ organoids and

kidney AML, compared to isogenic control organoids and to normal kidneys, respectively (Fig. 3f).

We next focused on enolase 2 (ENO2), a glycolysis enzyme upregulated in a variety of neuroendocrine tumors, including kidney AML (Supplementary Fig. 4d)[30–32]. We observed that *ENO2* mRNA expression was significantly higher in $TSC2^{-/-}$ organoids compared to $TSC2^{+/+}$ organoids (Fig. 3f), a result that

**Fig. 3 Gene expression analysis of $TSC2^{-/-}$, $TSC2^{+/-}$, and $TSC2^{+/+}$ renal organoids. a** Volcano plots showing the distribution of all differentially expressed genes (DEGs) in $TSC2^{-/-}$ renal organoids compared to $TSC2^{+/+}$ (left) and $TSC2^{+/-}$ (right) renal organoids (FDR < 0.05). Each dot represents a unique gene; red denotes $\log_2$ (fold change) >2, upregulated genes in $TSC2^{-/-}$; blue denotes $\log_2$ (fold change) <-2, downregulated in $TSC2^{-/-}$. Selected statistically significant upregulated and downregulated genes (NCBI/Entrez names) are indicated, as determined by a two-sided Chi-Square test. **b** Principal Component Analysis (PCA) of RNA-Seq data from renal organoids of the three genotypes, $n = 3$ samples for each genotype, five organoids *per* sample. **c** Heatmap showing hierarchical clustering of three different genotypes of kidney organoids using the top 3000 most variable genes. Color scale representative of gene expression level: red denotes $\log_2 \leq 3$, blue denotes $\log_2 \geq$ -3. **d** Representative enrichment plots corresponding to gene set enrichment analysis (GSEA) for pairwise comparison of $TSC2^{-/-}$ vs. $TSC2^{+/-}$. **e** Venn diagrams indicating 187 common differentially expressed genes, including signature AML markers, in $TSC2^{-/-}$ vs. $TSC2^{+/+}$ renal organoids and kidney AML *vs.* normal kidney. **f** Comparative mRNA expression levels for AML hallmark genes in $TSC2^{-/-}$, $TSC2^{+/+}$, and $TSC2^{+/-}$ renal organoids ($n = 3$ each) compared to human kidney AML ($n = 28$) and human kidney ($n = 8$). P values for individual comparisons done using a two-sided Mann–Whitney U test are indicated. Gene expression is shown in FPKM values. **g** Comparative *ENO2* mRNA expression levels in $TSC2^{+/+}$ and $TSC2^{+/-}$, $TSC2^{-/-}$ renal organoids ($n = 3$ each). P values for the indicated individual comparisons done using two-tailed Student's t test are shown. Gene expression is shown in FPKM values. **h**, **i** Box-and-whisker plot showing minimum value, first quartile, median, third quartile and maximum value for ENO2 content (**g**) and for enolase activity (**h**) in whole extracts of $TSC2^{+/+}$ and $TSC2^{-/-}$ renal organoids. P value for the 2-tailed Student's t test comparing $TSC2^{-/-}$ versus $TSC2^{+/+}$ is shown. $n = 4$ independent experiments, containing three organoids each.

was consistent with both increased ENO2 protein levels (Fig. 3g) and increased overall enolase activity in $TSC2^{-/-}$ organoids (Fig. 3g).

Taken together, the common patterns of expression for hallmark genes and signaling pathways further reinforced the similarities between $TSC2^{-/-}$ organoids and kidney AML, on a molecular level.

**$TSC2$ inactivation drives renal organoid tubule cyst formation**. Based on the dilated appearance of putative proximal tubules in $TSC2^{-/-}$ 3D organoids (Fig. 2a), we set out to study the phenotype of $TSC2^{-/-}$ nephrons. We noted that brightfield imaging of Day-21 2D $TSC2^{+/+}$ and $TSC2^{+/-}$ cell cultures contained tubule like structures, whereas cultures derived from $TSC2^{-/-}$ hiPSCs showed cavitated structures resembling cysts (Fig. 4a). Because cysts are another common abnormality found in TSC patients with renal manifestations[33], we investigated the cellular composition of the cysts to determine whether they were formed by tubule epithelial cells (TECs) derived from $TSC2^{-/-}$ hiPSCs. By means of immunofluorescence, we detected nephron segments corresponding to distal tubules expressing cadherin 1 (CDH1), proximal tubules containing brush borders labeled by lotus tetranoglobus lectin (LTL) and glomeruli containing podocytes expressing podocalyxin 1 (PODXL1) in the cell aggregates derived from $TSC2^{+/+}$ and $TSC2^{+/-}$ hiPSCs (Fig. 4b). In both cases, the segments looked anatomically normal and were sequentially connected, therefore indicating continuous nephrons (Fig. 4b). In the case of $TSC2^{-/-}$ cell aggregates, the cystic structures stained positive for both the CDH1 and LTL, indicating the cyst-like structures could comprise distal tubule and/or proximal tubule cells (Fig. 4c). Cyst formation occurred with a frequency of ~4.7 cysts/well for $TSC2^{-/-}$ cultures, compared with 0 and ~0.5 cysts/well for $TSC2^{+/+}$ and $TSC2^{+/-}$ cultures, respectively (Fig. 4b). Of note, 2D cyst-like structures developed despite the fact that *PKD1* expression without addition of cAMP modulators, such as forskolin (Fig. 4h), a method previously employed for the induction of cysts in kidney organoid models of polycystic kidney disease (PKD)[34]. Another observable phenotype in 2D $TSC2^{-/-}$ nephrons was that PODXL1$^+$ cells presented a less compact appearance, and were loosely grouped, suggesting dysmorphic glomerular structures (Fig. 4c). These phenotypic characteristics were not associated with changes in the expression of nephron segment markers such as *PODXL1*, *NPHS1*, and *SYNPO* (glomerulus); *AQP1* (proximal tubule); or *EPCAM* and *CDH1* (distal tubule), whose mRNAs were detected at comparable levels in our RNAseq analysis of $TSC2^{+/+}$, $TSC2^{+/-}$ and $TSC2^{-/-}$ organoids (Supplementary Fig. 5a), with exception of

*CDH2* (proximal tubule), which was upregulated in $TSC2^{-/-}$ organoids (Supplementary Fig. 5a).

These observations prompted us to analyze the spatial arrangements of cystic nephron structures in 3D organoids. On Day 18 of the 3D differentiation protocol, we observed protruding dome-like structures indicating incipient cyst formation on the surface of $TSC2^{-/-}$, but not of $TSC2^{+/+}$ or $TSC2^{+/-}$ differentiating spheroids by Day 16 (Supplementary Fig. 5b). On Day 21, individual cysts growing out of the $TSC2^{-/-}$ AML organoids were clearly observed by brightfield microscopy (Fig. 4d). Similar to what has been reported in organoid models of polycystic kidney disease, we found the size of cysts to be larger in 3D aggregates grown on low-attachment surfaces compared to 2D cell cultures (Supplementary Fig. 5b). Nephron segment analysis using immunofluorescence microscopy confirmed that $TSC2^{+/+}$ and $TSC2^{+/-}$ metanephric mesenchyme aggregates had formed kidney organoids as indicated by the detection of properly segmented nephrons in which CDH1$^+$ distal tubules were connected with LTL$^+$ proximal tubules, and PODXL1$^+$ nephrons, on Day 21 of the differentiation protocol (Fig. 4e). In $TSC2^{-/-}$ AML organoids, the analysis confirmed that the cells conforming the lining of the cysts stained positive for CDH1 and/or LTL, but not for PODXL1, indicating that similar to what is observed in TSC patients[35] the cysts are specifically associated with tubular nephron segments, but can also emerge from multiple segments (Fig. 4f, Supplementary Fig. 5c, d). The immunofluorescence analysis also showed that while mainly individual tubule segments were involved, cysts lined by a combination of both the proximal and distal tubule cellular components were also observed in $TSC2^{-/-}$ AML organoid cysts, possibly representing transition regions (Fig. 4f). Unlike the single-cell layers observed lining the cyst regions associated with well-defined tubule segments, the mixed-cell lining regions were multilayered (Fig. 4f), resembling the columnar epithelium that can be observed in the renal cysts of TSC patients[35]. The distribution of LTL staining, representing polarity of brush borders, revealed another aspect of TSC cystic disease, namely loss of cell polarity in the single-cell proximal tubule stretches of cyst lining, in contrast to the polarized cells observed in $TSC2^{+/-}$ tubules (Fig. 4g). Of note, cyst formation in $TSC2^{-/-}$ AML organoids did not involve loss of polycystin 1 (PKD-1) expression, as indicated by the similar mRNA levels detected in our organoid RNAseq analysis (Fig. 4h).

These findings indicated that $TSC2^{-/-}$ renal organoids recapitulate key features of cystic kidney disease associated with TSC, and support the concept that renal epithelial cyst formation is also driven by loss of *TSC2*.

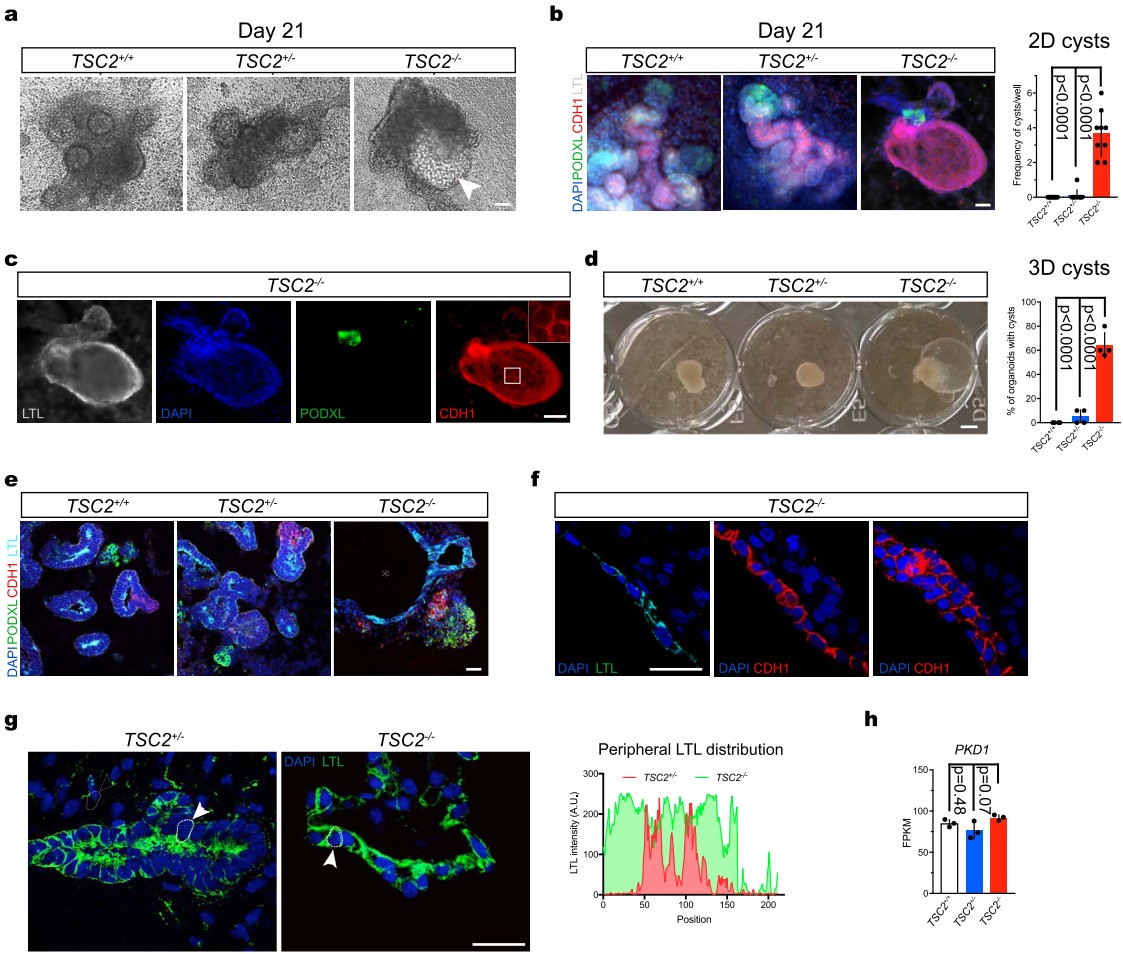

**Fig. 4 TSC2 inactivation drives cystogenesis during nephric differentiation. a** Representative brightfield images of 2D $TSC2^{-/-}$ hiPSC-derived renal tissues on Day 21 of differentiation. The arrowhead indicates a visible cyst. Twenty independent experiments were performed with similar results. **b** Immunofluorescence image showing 2D nephrons derived from the three hiPSC genotypes, labeled with PODXL1 (glomerulus), LTL (proximal tubule), and CDH1 (distal tubule). Number of cysts per well was quantified. Bar graphs represent mean ± SD, $n = 8$ wells from 2 independent experiments. **c** The cyst framed in the $TSC2^{-/-}$ nephron micrograph in (**a**) is stained for nephron markers by immunofluorescence with inset showing the distal tubule epithelium. Ten independent experiments were performed with similar results. **d** Brightfield image of 3D organoids of the three genotypes showing cystic $TSC2^{-/-}$ organoids, with quantification of cyst formation. Bar graphs represent mean ± SD, $n = 4$ experiments. **e** Representative confocal sections showing organoid glomeruli, proximal and distal tubule regions as indicated by PODXL1, LTL, and CDH1 staining. A cyst associated with a proximal tubule is shown in the $TSC2^{-/-}$ organoid section. Twelve independent experiments were performed with similar results. **f** Anatomical organization of $TSC2^{-/-}$ organoid cyst lining visualized using confocal imaging, both proximal tubule epithelial cell (PTEC) and distal tubule epithelial cell (DTEC) single-cell layers and DTEC multicellular regions are shown with LTL and CDH1 immunofluorescence. Eight independent experiments were performed with similar results. **g** Representative confocal imaging showing the alterations in polarity observed in PTECs lining $TSC2^{-/-}$ organoid cysts, compared to $TSC2^{+/-}$ organoid proximal tubule, using LTL staining. A representative quantification of LTL signal distribution in the periphery of individual cells is shown. Five independent experiments were performed with similar results. **h** Expression levels for *PKD1* in $TSC2^{-/-}$, $TSC2^{+/+}$, and $TSC2^{+/-}$ renal organoids. Bar graphs represent mean ± SD, $n = 3$ samples for each genotype, five organoids *per* sample. Gene expression is shown in FPKM values. *P* values for individual comparisons done using Tukey test in one-way ANOVA analysis are indicated. Scale bars, 50 µm (**a**, **b**, **c**, **e**), 25 µm (**f**, **g**), and 1 mm (**d**).

**Transplanted $TSC2^{-/-}$ AML organoids recapitulate TSC-associated kidney AML and cystic disease in vivo.** After determining that our in vitro $TSC2^{-/-}$ hiPSC-derived renal organoids recapitulated major features of AMLs TSC-associated cystic disease, we sought to investigate the effect of vascularization on phenotype development. The paucity of vascularization is a major limitation of in vitro hPSC-derived kidney organoid models[36]. We transplanted Day-18 $TSC2^{-/-}$ pre-organoid stage spheroids into the kidneys of 10-week-old immunodeficient RNU rats (Fig. 5a, b). We sought to establish how in vivo vascularization alters the anatomy and growth of $TSC2^{-/-}$ AML organoids, and whether this approach would result in an animal model capable of recapitulating TSC-associated renal abnormalities. Our strategy took advantage of the size of RNU rat kidneys, which allowed for

the simultaneous transplantation of two ~500 µm Day-18 spheroids (Fig. 5a, b). In addition, we transplanted organoids into both kidneys of each rat, giving us the ability to compare xenografts with different genotypes in the same animal.

Assessment of xenograft size indicated a significantly higher growth rate for $TSC2^{-/-}$ AML organoids (4.43 fold change), compared to $TSC2^{+/-}$ (3.55 fold change) or $TSC2^{+/+}$ kidney organoids (3.4 fold change) at Day 14 post-transplantation (Fig. 5c). At Day 14, cysts could be seen on the surface of kidneys containing $TSC2^{-/-}$ AML organoids but not in kidneys carrying the $TSC2^{+/-}$ or $TSC2^{+/+}$ organoids (Fig. 4d). At this time point, we surgically removed the rat kidneys and examined the xenografts. To distinguish human organoid tissues from rat tissues by means of immunofluorescence microscopy, we used

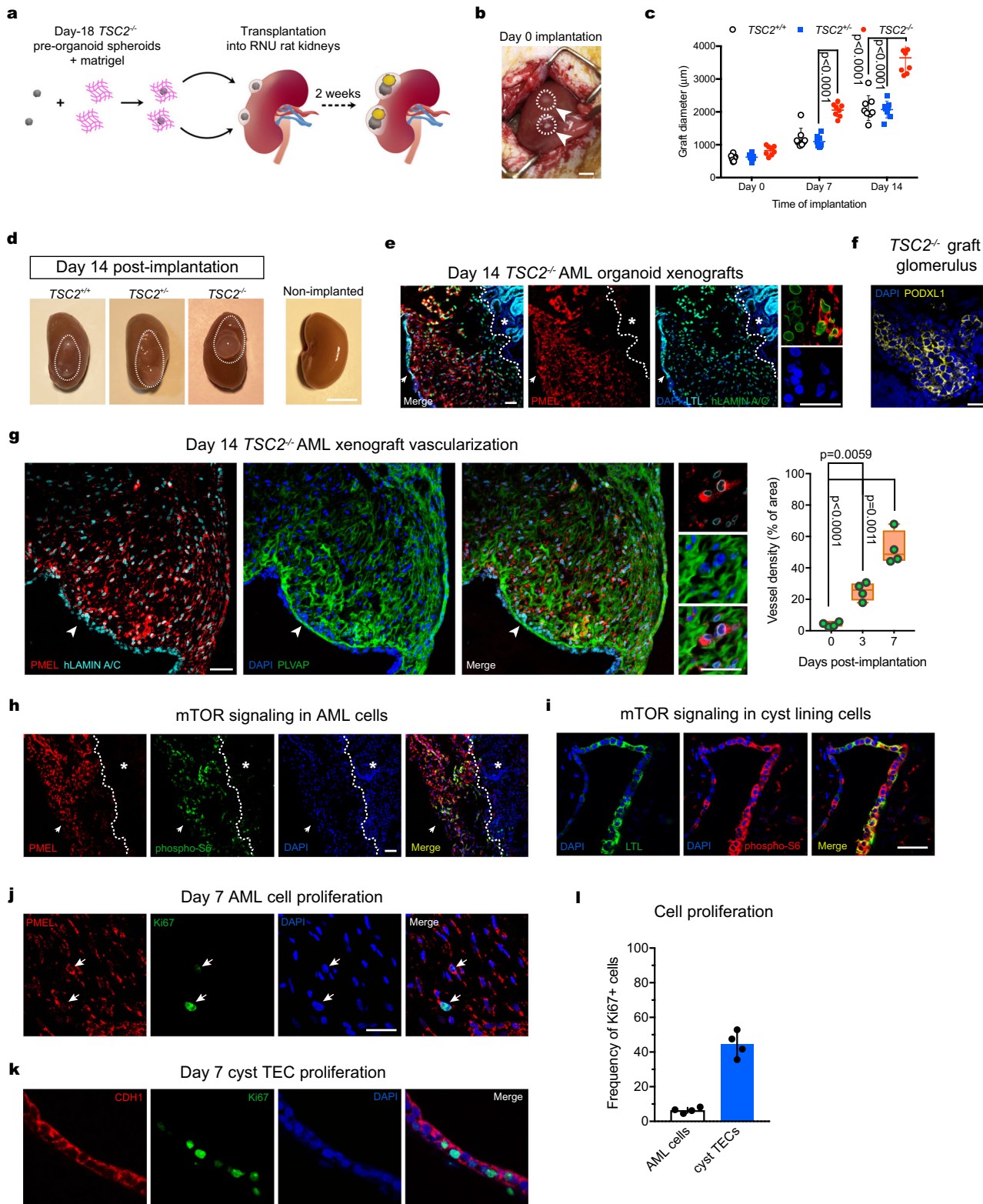

antibodies against either the human nuclear antigen (HNA) or the human isoform of Lamin A/C (hLamin A/C). Analysis of AML markers revealed the widespread and abundant presence of human cells expressing either ACTA2 or PMEL in the $TSC2^{-/-}$ xenografts, but not in $TSC2^{+/-}$ or $TSC2^{+/+}$ organoid xenografts, indicating that the AML phenotype was maintained after engraftment (Fig. 5e). In addition to AML-like cells, nephron tubule epithelium (Fig. 5e, arrowhead) and glomeruli (Fig. 5f)

were also preserved in the transplanted organoids. Of note, both the ACTA2$^+$ and PMEL$^+$ cells were observed in close proximity to microvascular endothelium of host origin that was visualized through detection of the marker plasmalemma vesicle-associated protein (PLVAP) (Fig. 5g), which is expressed by the micro-vasculature of the kidney[37]. In addition, mTOR activation, assessed by pS6 expression, was observed in both ACTA2$^{-/-}$ or PMEL$^+$ cells of $TSC2^{-/-}$ AML organoid graft cells, but not in the

**Fig. 5 Orthotopic $TSC2^{-/-}$ renal organoid xenografts model TSC-associated AML and TSC cystic disease in vivo. a** Schematic depicting the strategy used for the transplantation of Day 18 pre-organoid spheroids in the subcapsular region of RNU rat kidneys. **b** Representative image showing two spheroids implanted on Day 0, each indicated by an arrowhead. **c** Quantification of graft size over the period of two weeks. Scatter dot plot shows mean ± SD, $n = 8$ grafts, 4 rat kidneys. $P$ values determined by two-way ANOVA analysis using Tukey test for multiple comparisons are indicated. **d** Representative image of RNU rat kidneys with engrafted $TSC2^{+/+}$, $TSC2^{-/-}$, and $TSC2^{+/-}$ organoids on Day 14 post-transplantation. **e** Representative confocal sections of $TSC2^{-/-}$ AML grafts showing wide distribution of human PMEL$^+$ cells and LTL$^+$ PTECs forming the cyst lining, the latter indicated by a white arrow. LTL-stained rat proximal tubules are indicated by a white star. Eight independent experiments were performed with similar results. **f** Representative confocal image of a glomerulus containing labeled with PODXL1$^+$ cells in a $TSC2^{-/-}$ AML graft. Eight independent experiments were performed with similar results. **g** Representative immunostaining and confocal images showing the thorough vascularization indicated by PLVAP$^+$ vessels, observed in close apposition to PMEL$^+$ cells. Quantification of vessel density shown in the floating bar graph for grafts harvested on Day 0, Day 3 and Day 7 post-transplantation. Values represent mean ± SD, $n = 4$ grafts. $P$ values for individual comparisons done using Tukey test in one-way ANOVA analysis are indicated. **h, i** Representative confocal immunofluorescence images for phospho-S6 in PMEL$^+$ cells (**h**) and cystlining PTECs (**i**) in $TSC2^{-/-}$ AML xenografts. In (**h**) the white arrow indicates the luminal compartment, while the white star indicates the rat kidney. Five independent experiments were performed with similar results. **j, k** Detection of Ki67 in PMEL$^+$ cells (**j**) and PTECs (**k**) in $TSC2^{-/-}$ AML grafts on Day 7 post-transplantation. The white arrows indicate Ki67-expressing human PMEL$^+$ cells. Five independent experiments were performed with similar results. **l** Quantification of PMEL$^+$ Ki67$^+$ and LTL$^+$ Ki67$^+$ cells. Values in the bar graph represents mean ± SD, $n = 4$ grafts. Scale bars, 1 cm (**b**), 2 cm (**d**), 50 μm (**e, f, g, h, i**), 25 μm (**e, g** high magnification panels), 25 μm (**i, j, k**).

adjacent normal rat tissue or in $TSC2^{+/+}$ or $TSC2^{+/-}$ organoid xenografts, indicating that metabolic activation was consistent with xenograft growth (Fig. 5h). Analysis of the tubule epithelium in $TSC2^{-/-}$ xenografts confirmed the formation of cystic tubules that were lined by PTECs and DTECs in varying proportions, arranged into stretches of single-cell lining, that alternated with stretches presenting a multilayered organization, as observed in $TSC2^{-/-}$ organoids in vitro (Fig. 4f). Similar to the TECs found in the in vitro cysts of $TSC2^{-/-}$ AML organoids, the TECs present in the in graft cysts lacked polarity and had activated mTORC1 signaling. Similar to the AML cells, activation of mTORC1 activation was also observed in epithelial cyst cells of the $TSC2^{-/-}$ grafts (Fig. 5i). Despite the common activation of mTORC1 observed in both the AML cells and TECs, most proliferative activity occurred in the cysts lining (Fig. 5j, k, l), suggesting that graft growth was mainly a consequence of cyst hyperplasia. This result is consistent with both the lack of proliferative activity observed in classic kidney AML cells[38], and the neoplastic activity observed in TSC-associated kidney cyst lining epithelial cells[35,39].

Collectively our in vivo experiments showed that orthotopic transplantation enhanced the phenotype of $TSC2^{-/-}$ AML organoids, permitting the recapitulation of human TSC-associated AML in vivo with a high degree of anatomical and molecular fidelity.

**Delivery of rapamycin-loaded nanoparticles abrogates orthotopic $TSC2^{-/-}$ AML xenografts.** Our results with transplanted $TSC2^{-/-}$ AML organoids in RNU rats prompted us to use this animal model to examine the effect of a nanoparticle formulation for delivery of the mTOR inhibitor rapamycin, a compound used for the treatment of AML tumors in TSC patients[33]. To this end, we designed an experiment in which Rapa was encapsulated in ultra-fine solid biopolymeric nanoparticles and delivered into the subcapsular space adjacent to the organoid grafts (Supplementary Fig. 6a–e). We devised this approach to enable targeted delivery of small doses of rapamycin locally over systemic delivery, which can cause undesired drug effects in other organs. We confirmed the absorption of rapamycin-loaded nanoparticles by kidney organoids in vitro, by visualizing the uptake of nanoparticles loaded with Bodipy 650/665-X (Supplementary Fig. 6f). Incubation of kidney organoids with nanoparticles loaded with 500 ng rapamycin, in vitro, resulted in progressive loss of tissue viability within 48 h (Supplementary Fig. 6g). Based on these results, we tested two doses of rapamycin nanoparticles (Rapa-Nps) for the in vivo treatments, namely 500 ng and 2 μg, which we injected under the kidney capsule approximately 2 mm away from each

$TSC2^{-/-}$ AML organoid xenograft, 2 weeks after transplantation. We repeated the treatment 3 days later to complete a regimen of two Rapa-Np injections, and harvested the whole kidneys one week after the first injection (Fig. 6a). We measured the size of the organoid xenografts 3 days and 1 week, i.e., at tissue-collection time- after the first injection. Our measurements showed that Rapa-Np treatment resulted in a rapid reduction of organoid xenograft size (Fig. 6b) of cyst size, with only a few scattered cystlining cells containing pyknotic nuclei observed at the end of the treatment (Supplementary Fig. 7a). This result was in contrast with the effect of oral rapamycin, which effectively blocked xenograft growth but did not result in significant shrinkage after 14 days of treatment (Supplementary Fig. 7b). Immunofluorescence staining showing Caspase 3 (CASP3) activation in PMEL$^+$ cells of treated xenografts, but not in cells of control nontreated xenografts, suggested that the Rapa-Np treatment-induced cell apoptosis (Fig. 6c, Supplementary Fig. 7c). Western blotting of organoid xenografts showed increased cleaved caspase 3 at Days 3 and 7 of treatment (Fig. 6e). In addition to the analysis of CASP3 activity, we detected increasing DNA fragmentation in PMEL$^+$ cells of treated xenografts by means histological analysis of terminal deoxynucleotidyl transferase dUTP nick end labeling (TUNEL) (Fig. 6e). DNA fragmentation was not observed in control nontreated $TSC2^{-/-}$ AML organoid xenografts (Fig. 6e, Supplementary Fig. 7d). Both CASP3 activity and DNA fragmentation indicated that delivery of Rapa-Nps was an effective treatment for local ablation of $TSC2^{-/-}$ AML organoid xenografts.

Taken together, these results highlight the $TSC2^{-/-}$ AML organoid xenografts in RNU rats as a powerful preclinical model for testing drugs and therapeutic approaches for the treatment of AMLs.

**Rapamycin nanoparticles disrupt the interaction between p21$^{CIP1}$ and pro-CASP3 in $TSC2^{-/-}$ AML organoid xenografts.** Current oral rapalog therapies only partially shrink kidney AMLs in TSC patients[8,9]. Whether mechanisms of tumor resistance reduce the efficacy of these treatments remains unknown. To answer this question we looked at the expression of genes involved in tumor resistance and identified *CDKN1A* as a gene overexpressed in both $TSC2^{-/-}$ AML organoids and kidney AML compared to control organoids and to normal kidney (Fig. 7a). Further analysis using The Cancer Genome Atlas indicated that *CDKN1A* is significantly upregulated in kidney AML compared to other tumors (Fig. 7b). These results led us to explore a previously described role for *CDKN1A* as a driver of tumor resistance through stabilization and accumulation of its encoded protein,

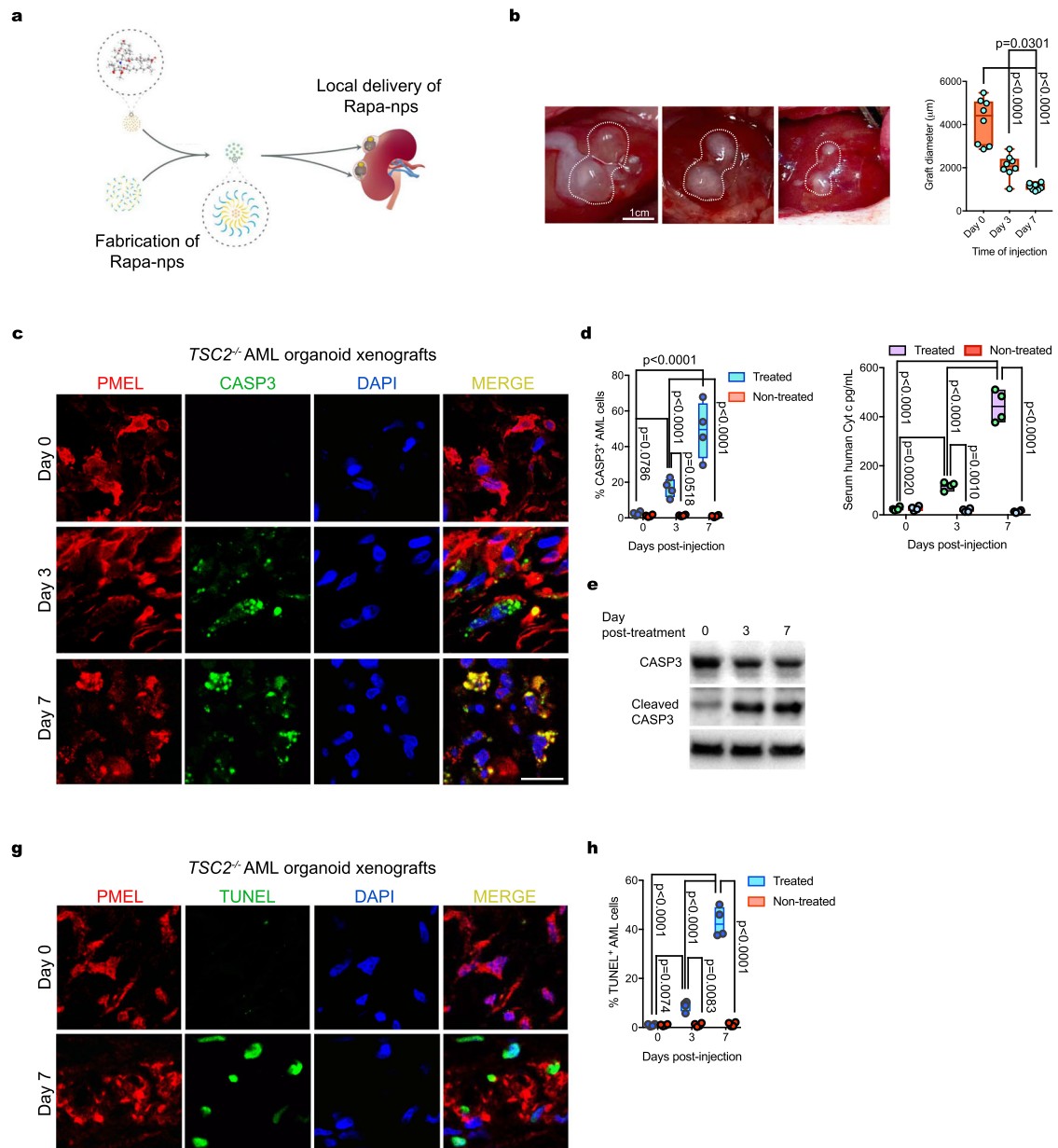

**Fig. 6 Ablation of TSC2$^{-/-}$ AML organoid xenografts treated with Rapamycin-loaded nanoparticles. a** Schematic showing the strategy for the delivery of Rapa-nanoparticles locally, near the TSC2$^{-/-}$ AML organoid xenografts. **b** Representative photographs showing the size of TSC2$^{-/-}$ xenografts on Day 3 and Day 7 post Rapa-Np delivery. Quantified diameter values on the floating bar graph represent mean ± SD, n = 8 grafts. P values for individual comparisons done using Tukey test in one-way ANOVA analysis are indicated. **c** Representative immunofluorescence images for the detection of activated Casp3 in TSC2$^{-/-}$ AML organoid xenograft PMEL$^+$ myoid cells on Day 3 and Day 7 postdrug delivery. Five independent experiments were performed with similar results. **d** Floating bar graph with quantification of PMEL$^+$ Casp3$^+$ cells on Days 3 and 7 postdrug delivery. Mean ± SD are reported, n = 4 grafts. P values determined by two-way ANOVA analysis using Tukey test for multiple comparisons are indicated. **e** Detection of cleaved Casp3 on Day 0 (free organoid), 3 and 7 in protein extracts from TSC2$^{-/-}$ AML xenografts (Day 3 and 7). **f** Quantification of human Cytochrome C in the serum of RNU carrying rats TSC2$^{-/-}$ AML organoid xenografts, treated with Rapa-Np and controls that did not receive the treatment. Floating bar graph shows mean ± SD, n = 4 rats per group. P values determined by two-way ANOVA analysis using Tukey test for multiple comparisons are indicated. **g** Representative confocal imaging depicting detection of DNA fragmentation by TUNEL on sections of TSC2$^{-/-}$ AML xenografts, three and seven days post Rapa-Np delivery. Five independent experiments were performed with similar results. **h** Floating bar graph for the quantification of PMEL$^+$ TUNEL$^+$ cells shows mean ± SD are reported, n = 4 grafts per group. P values determined by two-way ANOVA analysis using Tukey test for multiple comparisons are indicated. Scale bars, 1 cm (**b**), 25 mm (**c**, **g**).

namely p21$^{CIP1}$, in the cell cytoplasm[40,41]. We analyzed p21$^{CIP1}$ distribution and observed protein accumulation in the cytoplasm of both the TSC2$^{-/-}$ AML organoids and kidney AML (Fig. 7c). This result was consistent with increased accumulation of p21$^{CIP1}$ in TSC2$^{-/-}$ AML organoid xenografts compared to control

TSC2$^{+/+}$ iPSC-derived xenografts (Fig. 7d), a finding that was concomitant with higher levels of pro-CASP3 in TSC2$^{-/-}$ AML organoid xenografts (Fig. 7d). Rapa-Np treatment reduced the levels of p21$^{CIP1}$ in treated TSC2$^{-/-}$ AML organoid xenografts, compared to nontreated ones, while simultaneously triggering

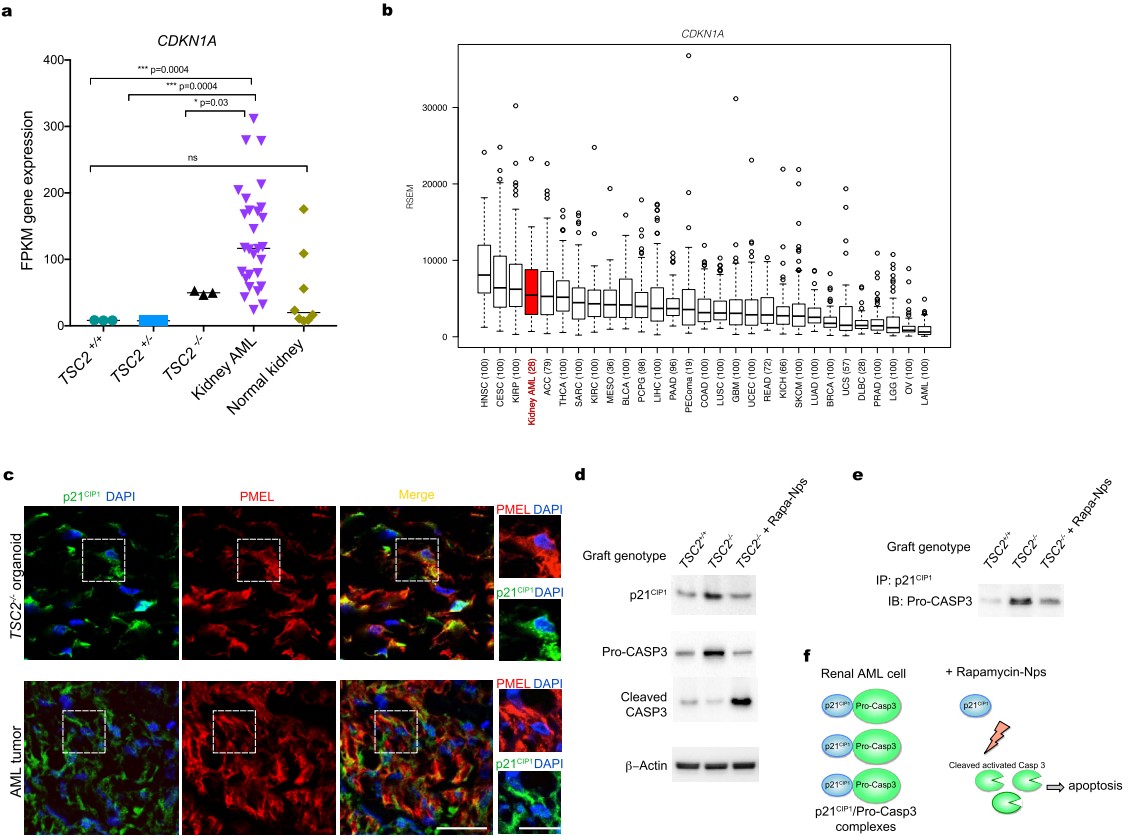

**Fig. 7 Rapamycin-Nps disrupt the interaction between p21[CIP1] and pro-CASP3 in TSC2[−/−] organoid AML cells in vivo. a** RNAseq data box plots showing expression levels of *CDKN1A* in *TSC2[−/−]*, *TSC2[+/−]* and *TSC2[+/+]* renal organoids (*n* = 3 each, five organoids *per* ample), compared to human kidney AML (*n* = 28) and human kidney (*n* = 8) samples. *P* values for individual comparisons done using a two-sided Mann–Whitney U test are indicated. Gene expression is shown in FPKM values. **b** Box-and-whisker plot showing minimum value, first quartile, median, third quartile and maximum value for expression of *CDKN1A* in various tumors using The Cancer Genome Atlas data. Kidney AML is highlighted in red. Sample size for each type of tumor: HNSC *n* = 100, CESC *n* = 100, KIRP *n* = 100, kidney AML *n* = 28, ACC *n* = 79, THCA *n* = 100, SARC *n* = 100, KIRC *n* = 100, MESO *n* = 36, BLCA *n* = 100, PCPG *n* = 98, LIHC *n* = 100, PAAD *n* = 96, PEComa (19), COAD *n* = 100, LUSC *n* = 100, GBM *n* = 100, UCEC *n* = 100, READ *n* = 72, KICH *n* = 66, SKCM *n* = 100, LUAD *n* = 100, BRCA *n* = 100, UCS *n* = 57, DLBC *n* = 28, PRAD *n* = 100, LGG *n* = 100, OV *n* = 100, LAML *n* = 100. **c** Representative immunostaining and confocal images showing PMEL and p21[CIP1] in *TSC2[−/−]* AML organoids and in kidney AML tumor samples. High magnifications show cytoplasmic p21[CIP1] signal. Three independent experiments were performed with similar results. Scale bars, 25 μm, and 12.5 μm for high magnification panels. **d** Representative Western Blots showing levels of p21[CIP1], pro-CASP3 and CASP3 in transplanted *TSC2[+/+]* and *TSC2[−/−]* organoids. Three independent experiments were performed with similar results. **e** Representative immunoblots showing co-immunoprecipitated p21[CIP1] and pro-CASP3 in transplanted *TSC2[+/+]* and in *TSC2[−/−]* renal organoids untreated or treated with Rapa-Nps. Three independent experiments were performed with similar results.

CASP3 cleavage, by Day 7 (Fig. 7d). The concomitant loss of p21[CIP1] and activation of CASP3 prompted us to investigate the prospective interaction between the p21[CIP1] and pro-CASP3. By means of the co-immunoprecipitation assay we determined that p21[CIP1] directly interacts with pro-CASP3 in *TSC2[−/−]* AML organoid xenografts, and that this interaction is abrogated by treatment with Rapa-Nps, resulting in accumulation of cleaved CASP3 (Fig. 7e).

Collectively, these data validated our *TSC2[−/−]* iPSC-derived AML organoid xenograft model as an effective tool to investigate mechanisms of AML with implications for the design of more effective therapeutic strategies.

## Discussion

Here we report an experimental strategy for the generation of in vivo models of human renal diseases associated with rare genetic disorders that cannot be reproduced in all their complexity with current experimental animal models. Using a combination of genome editing and hiPSC techniques we show that nephric differentiation of hiPSCs carrying biallelic LOF mutations in the *TSC2* locus generates renal tissues that reproduce anatomical and molecular aspects of human myoid AML tumors.

The AML identity of PMEL[+], ACTA2[+] *TSC2[−/−]* organoid myoid cells is supported by two critical findings, namely the dual smooth muscle and melanocyte cell phenotype, and cell morphology[5,6]. Melanocytes do not express ACTA2, as indicated by single-cell transcriptome profiles of skin cells[42,43]. Similarly, mesenchymal cells in normal kidney tissues do not express melanocyte markers. Both our flow cytometry and histology results show that ACTA2[+] mesenchymal cells present in *TSC2[+/−]* or *TSC2[+/+]* kidney organoids did not express melanocyte genes, a result that is consistent with single-cell transcriptome profiles of WT kidney organoid mesenchymal cells[43]. Last, in terms of morphology, *TSC2[−/−]* organoid AML cells show a distinct plump myoid morphology with that is characteristically observed in AML tumor cells[6,44], and which clearly differs from the morphology of kidney fibroblasts characterized by elongated spindle-shaped cell bodies ending in prolonged processes, as well as from the multipolar dendritic morphology of skin melanocytes[25].

$TSC2^{-/-}$ AML organoids also recapitulated the epithelial cysts observed in TSC patients. Cystic kidney disease is the second most frequent renal manifestation of TSC and can be triggered by the biallelic inactivation of $TSC1/TSC2$ alone[39,45] or as the result of deletion events that also affect the $PKD1$ locus located adjacently in chromosome 16[46]. Our RNAseq data showing that levels of $PKD1$ mRNA in $TSC2^{-/-}$ AML organoids were comparable to mRNA levels in $TSC^{+/-}$ and $TSC^{+/+}$ kidney organoids, indicated that the cystogenic mechanisms are solely driven by the loss of $TSC2$. Cyst formation can also be associated with a subtype of AML called AML containing epithelial cysts (AMLECs) that can be surrounded by condensed subepithelial cambium-like AML cells[47,48]. In the case of AMLECs, the mechanisms of cystogenesis are not well understood, and a role for AML cells, possibly through the secretion of paracrine factors, has been suggested[47]. While those mechanisms have not been investigated in this study, the presence of both an AML phenotype and epithelial cysts is congruent with the idea that early developmental LOH events affect a broader spectrum of kidney tissues, therefore resulting in multiple RAs[14,33].

Our experiments in RNU rats represent a major step toward modeling TSC-associated renal manifestations in vivo using transplanted hiPSC-derived kidney organoids. The rat kidney critically provided a vascular bed supporting the growth of $TSC2^{-/-}$ organoids, promoting the hyperproliferative activity of cyst lining epithelial cysts, consistent with the clinical notion that factors present in the kidney and the blood play an important role in the development of cystic kidney disease. This in vivo organoid strategy can increase the throughput of disease modeling studies by supporting multiple organoid xenografts that can be used for drug-testing and for mechanistic interrogation. We combined in vitro and in vivo experimental settings to identify over-expression of $CDKN1A$ and direct interaction between its encoded protein p21$^{CIP1}$ and pro-CASP3 as a mechanism that can be overridden by nanocarrier-based strategies designed to increase rapalog bioavailability at the site of the lesions. These results are consistent with a previously reported role for p21$^{CIP1}$ in tumor resistance[40,41], and with the observation that in certain tumors, accumulation p21$^{CIP1}$ in the cytoplasm is associated with tumor survival and reduced tumor sensitivity to treatment[49–54]. These findings will be relevant for the design of oral nanotherapies with increased antitumor efficacy. To conclude, we expect that pre-clinical animal models carrying transplanted hiPSC-derived organoids may become a widely used experimental platform for the study of disease mechanisms and for the development of therapies against molecular targets in human kidney tissues.

## Methods

**hiPSC line maintenance**. The hiPSC cell lines 77-patient, 77-TSC2-null, and control 77-TSC2WT were a kind gift from Dr. Sahin. The 77-patient line carrying a heterozygous 9-bp deletion in the $TSC2$ locus was derived from the lymphoblasts of a patient with TSC using the sendai-virus method for the delivery of OCT3/4, KLF4, L-MYC, SOX2, and LIN28 into the cells[19]. The 77-TSC2-null line was generated by introducing a second microdeletion of 6-bp in the WT allele of the 77-patient line, using transcription activator-like effector nucleases TALEN[19]. The 77-TSC2WT line was created by correcting the heterozygous microdeletion of TSC2 in 77-patient cell line with CRISPR-cas9 method[19]. The three hiPSC lines were maintained in mTeSR™1 medium (STEMCELL Technologies # 85850), in 6-well tissue culture plates (Falcon, #353046) coated with 1% vol/vol LDEV-Free hESC-qualified Matrigel (Corning Life Sciences # 354277) in a 37 °C incubator with 5% CO2. Cells were passaged at a 1:3 split ratio once a week, using Gentle Cell Dissociation Reagent (GCDR, STEMCELL Technologies #07174). Each cell line was maintained below passage 45 and mycoplasma contamination was not detected. Permission for the use of hiPSC lines was granted to DRL by our institutional review board (IRB) at Brigham and Women's Hospital, through IBC protocol 2020B000020.

**hiPSC differentiation**. hiPSCs maintenance cultures were washed once with PBS (Life Technologies, #10010-049) and dissociated into single-cell suspensions using Accutase (STEMCELL Technologies, #07920). One hundred and fifty thousand cells were plated onto 24-well tissue culture plates (TPP, #92024) coated with 1% Matrigel in mTeSR™1 medium supplemented with ROCK inhibitor molecule Y27632 (10µM) (TOCRIS, #1254). After 24 h, the mTeSR™1 medium was replaced by basic differentiation medium consisting of Advanced RPMI 1640 (Life Technologies, #12633-020) containing 1X L-GlutaMAX (Life Technologies, #35050-061) supplemented with 8 µM CHIR99021 (Sigma-Aldrich, #SML1046) for 4 days. Cells were then cultured in Advanced RPMI + 1X L-GlutaMAX + Activin A (10 ng/ml) (R&D, #338-AC-050) for 3 days. For induction of metanephric mesenchyme the media was replaced with Advanced RPMI + 1X L-GlutaMAX + FGF9 (20 ng/ml) (R&D, #273-F9-025/CF) for 7 days, with a 1-h pulse of 3µM CHIR added on day 9. To generate 3D organoids, the cells were lifted using Accutase on Day 9 and the cell suspensions incubated with CHIR pulse medium 37 °C, 5% CO2 for 1 h then switched to basic differentiation medium + FGF9 and plated at 50,000 cells per well onto ultra-low-attachment plates (Corning, #7007). The plates were centrifuged at 1,500 r.p.m. for 15 s, and the cells then cultured 37 °C, 5% CO2 until day 13. At day 13, a pulse of 1µM retinoic acid was added to the medium containing FGF9, and the organoid or cell cultures were incubated for 24 h. At day 14, the cultures where transferred to basic differentiation medium with no additional factors for 7–14 days, and harvested at day 21-28. The differentiation reagents and procedures can be accessed via Protocol Exchange (https://protocolexchange.researchsquare.com), using DOI: 10.21203/rs.3.pex-1648/v1.

**Induction of organoid fibrosis with IL-1β**. Day-21 $TSC2^{+/+}$ organoids were incubated in Advanced RPMI + 1X L-GlutaMAX containing 50 ng/ml human IL1-β (Sigma-Aldrich # H6291) for 96hs[29]. The medium containing IL1-β was replaced every 48 h.

**Quantitative RT-PCR**. Total RNA was extracted using the RNeasy Plus MiniKit (Qiagen). Purity was determined byA260–A280. cDNA was synthesized using oligo(dT) and random primers (iScript Reverse Transcription Supermix; Biorad). Quantitative PCR was performed using the QuantStudio 7 Flex Real-Time PCR System (ThermoFisher Scientific) using TaqMan Gene Expression Assays (ThermoFisher Scientific #4331182). The specific primer pairs used in quantitative PCR are listed in Supplemental Table 1.

**Western blot**. In vitro and grafted organoids were lysed in RIPA buffer containing Halt Protease and Phosphatase Inhibitor Cocktail (Thermo FisherScientific), using a glass Dounce homogenizer (DWK Life Sciences Kimble™) on ice. Following protein quantification, SDS-PAGE gel run and western blotting were performed. The following antibodies from Cell Signaling Technology were used: antiribosomal protein S6 (#2217, clone 5G10, 1:1000), antiphospho S6 (#2211, 1:1000), anti-phospho P70S6K (#9205, 1:1000), anti-P70S6K (#9202, 1:2000), anti-β Actin (#4967, 1:1000), anti-GPNMB (#38313, clone E4D7P, 1:700). Anti-ACTA2 was from Sigma-Aldrich (#F3777, clone 1A4, 1:2000), antihuman pro-Caspase 3 was purchased from ThermoFisher Scientific (#MA1-41163, clone 31A893, 1:400), antihuman cleaved Caspase 3 was purchased from Abcam (ab2302, 1:200), antihuman p21CIP1 was purchased from Novus Biologicals (#AF1047, 1:400). Primary antibodies were detected with peroxidase-conjugated antirabbit or anti-mouse IgG (1:3000) and visualized with SuperSignal West Femto Substrate (ThermoFisher Scientific #34094). Blots were quantified using ImageJ v1.51 W (https://imagej.nih.gov/ij/).

**Immunofluorescence**. Organoids and grafts were fixed with 4% paraformaldehyde in PBS for 30 min in a 96-well plate, washed three times in PBS, then incubated with 30% sucrose (w/w) overnight at 4 °C. The next day the tissues were mounted into frozen blocks with O.C.T. compound (Fisher Scientific, #23-730-571) and were cut into 12-µm sections onto glass slides (Fisher Scientific #22-037-247). The sections were washed three times for 5 min in PBS, then incubated in blocking buffer containing 0.15% Triton X-100 and 5% normal donkey serum) for 1 h, followed by incubation with primary antibodies in antibody dilution buffer (0.15% Triton X-100 and 1% BSA in PBS) for 2 h or overnight, then washed three times in PBS. Incubation with secondary antibodies in antibody dilution buffer was performed for 1 h, followed by three washes in PBS and coverslips mounted on Vectashield with DAPI (Vector Labs #H-1200). The primary antibodies used were: anti-SIX2 (Proteintech, #11562-1-AP, 1:250), anti-SALL1 (Abcam #ab41974, clone K9814, 1:100), anti-PAX8 (Proteintech, #10336-1-AP, 1:250), anti-PMEL (Agilent Technologies, #M063429-2, clone HMB-45, 1:200), anti-ACTA2 (Sigma-Aldrich #C6198, clone 1A4, 1:400), anti-GPNMB (Cell Signaling Technologies #38313, clone E4D7P, 1:300), anti-PODXL1 (R&D Systems, #AF1658, 1:400), anti-CDH1 (Abcam # AB40772, clone EP700Y, 1:400), antihuman Lamin A/C (Abcam, #ab108595, clone EPR4100, 1:300), anti-PLVAP (Univ. of Iowa, MECA32, clone SP2/0, 1:200), antiphospho S6 (CST #2211, 1:400), anti-Ki67 (ThermoFisher Scientific, # 701198, clone 12H15 L5, 1:250), antihuman cleaved Caspase 3 (Cell Signaling Technology #9661, 1:300). Biotinylated lotus tetranoglobus lectin was purchased from Fisher Scientific (#NC0370187, 1:400) and visualized with streptavidin-Cy5 (ThermoFisher Scientific, #SA1011, 1:700). Images were taken using a Nikon C1 confocal microscope.

**Fabrication of rapamycin nanoparticles and gavage delivery**. Rapamycin was purchased from LC Laboratories (# R5000), Poly(ethylene glycol)-block-poly($\varepsilon$ – caprolactone) methyl ether (PEG-PCL, PCL average Mn ~5000, PEG average Mn ~5000) and Tetrahydrofuran (THF, ≥ 99.9%) were acquired from Sigma-Aldrich. Bodipy (FL and 650/665-X, SE) were obtained from Invitrogen. A solvent evaporation method under ultrasonication was used for the encapsulation of rapamycin with PEG-PCL micelles. A 20:1 theoretical loading ratio was used for the system fabrication (10 mg of PEG-PCL diblock copolymer and 0.5 mg of rapamycin). Rapamycin and PEG-PCL were dissolved in tetrahydrofuran and added drop by drop into MillQ water under ultrasonication (Sonic Dismembrator Model 500, Fisher Scientific, Waltham, MA) at 20% amplification. The solvent was left to evaporate overnight, after which loaded micelles nanoparticles were filtered through a 0.2 um syringe filter to remove non-encapsulated aggregates in the solution. The solution was then centrifuged for 20 min at a rotational speed of 1150 × $g$ at room temperature. The preparation was stored at 4 °C to avoid degradation. Nanoparticle morphology and size was analyzed using a JEOL 2010 Advanced High-Performance Transmission Electron Microscope (JEOL, MIT MRSEC, MA). The hydrodynamic radius and Zeta potential for the fabricated rapamycin-loaded nanoparticles were determined by photon correlation spectroscopy using a Zetasizer Nano ZS (Malvern Instruments, Worcester-shire, UK). The hydrodynamic radius was measured using a polystyrene cuvette, and the Zeta potential using a folded capillary Zeta cell. Hydrodynamic radius is reported in nm and Zeta potential in mV. Chemical analysis of functional groups of the nanoparticles was done using an FTIR spectrophotometer. Measurements for the infrared spectra of the nanoparticles were studied comparing the spectra of the reactants before and after the fabrication to observe the change in chemical composition. Daily oral delivery of rapamycin was done by gavage, at a concentration of 0.5 mg/Kg in DMSO.

**Organoid transplantation and nanoparticle delivery**. Ten-week-old male NIH-Foxn1$^{rnu}$ immunodeficient nude rats were purchased from Charles River Laboratories and maintained in the animal unit located at the Harvard Institutes of Medicine. For the surgery, the animals were anesthetized using isoflurane and weighed. The kidney was reached through a 2-cm retroperitoneal incision, and two 18-day organoids embedded in LDEV-Free hESC-qualified Matrigel were placed in the subcapsular space through a 2-mm incision in the kidney capsule. After the dorsal incision was sutured, the animals received 1 ml saline and allowed to recover. Two weeks after organoid transplantation, 2 µl of rapamycin-loaded nanoparticle suspensions were mixed with 10µl Matrigel and injected under the kidney capsule, approximately 2-mm away fom each organoid graft. All experiments were performed under animal protocol 2018N00066, approved by the Institutional Animal Care and Use Committee at Brigham and Women's Hospital. We have complied with all relevant ethical regulations for animal testing and research.

**Flow cytometry**. In each experiment, five organoids of each genotype were incubated with TrypLE Select (Thermo Fisher Scientific #12563011) in a shaker at 37 °C for 10 min. The digestions were passed through a 40-µm cell strainer (Millipore Sigma #CLS431750), and washed with PBS. Resulting single-cell suspensions were collected by centrifugation at 400 $g$ for 5 min. Intracellular stainings with anti-PMEL antibody HMB-45-FITC (Novus Biologicals #NBP2-34638F, 1:200) and with anti-ACTA2 (Sigma-Aldrich #C6198, 1:400) were performed using the PerFix-nc Kit (Beckman Coulter #B31167), for 30 min at 4 °C in supplemented PBS containing 2 mM EDTA and 2% FBS. Staining for human GPNMB (Fisher Scientific #17-983-842) was done on nonfixed live cell suspensions for 30 min at 4 °C in supplemented PBS containing 2 mM EDTA and 2% FBS. Analysis was performed on LSRII (Becton Dickinson) equipped with three lasers. Data were collected using FacsDIVA v4.1 software. Sorting gates were defined on the basis of fluorescence-minus-one control stains, using FlowJo 10.4.1.

**EdU TUNEL imaging**. TUNEL imaging was performed on frozen fixed graft tissue sections using the Click-iT TUNEL Alexa Fluor Imaging Assay (Invitrogen), following the manufacturer's instructions. Briefly, the sections were incubated with anti-PMEL antibody for 1 h at room temperature, then washed and fixed with 4% PFA for 15 min at room temperature, followed by permeabilization with 0.25% Triton X-100 in PBS for 20 min at room temperature. The sections were then incubated for 1 h at 37 °C with labeling reaction mix containing TdT and nucleotide mix including EdUTP. The sections were washed and incubated in Click reaction mix including the Alexa Fluor 488 azide. After washing, the slides were incubated with Alexa 546 anti-mouse secondary antibody and stained with DAPI.

**Whole organoid transcriptome RNA sequencing and bioinformatics analysis**. For each sample, total RNA was extracted from 5 organoids using the RNeasy Plus MiniKit (Qiagen). Purity was determined byA260–A280. Copy DNA (cDNA) library preparation and sequencing was performed at the Molecular Biology Core Facilities at Dan- Farber Cancer Institute. Samples were sequenced using an Illumina MiSeq sequencer and paired-end 100-bp reads generating 20–30 M reads *per* sample library. Raw Illumina output was converted to Fastq format using Illumina Bcl2fastq v2.18. The hg19 reference sequence was used for sample alignment.

DESeq2 differential gene expression analysis was done using Qlucore v3.6. Hallmark, KEGG and Gene Ontology (GO) analyses for organoid comparisons were performed using Gene Set Enrichment Analysis tool (GSEA; https://www.gsea-msigdb.org) using a phenotype permutation of 1000. The false discovery rate (FDR) used to define overrepresented gene sets was $q < 0.25$.

**Human kidney angiomyolipoma RNA sequencing**. Whole RNA samples were prepared from 18 fresh/frozen kidney AML samples. Copy DNA (cDNA) library preparation and sequencing was performed at the Molecular Biology Core Facilities at Dan- Farber Cancer Institute. Samples were sequenced using an Illumina HiSeq300 sequencer. All samples were discard pathologic specimens derived from patients who underwent surgical tumor resection. The samples were anonymized, no information regarding gender, age, or genotype was available. This set of samples were combined with publicly available data from 10 additional kidney angiomyolipomas and four normal kidneys[55] for downstream analyses. The protocol 2014P001446 "Discard human pathologic specimens" was approved by the Partners Human Research Committee of the Mass General Brigham hospitals.

**Enolase 2 and Cytochrome C ELISA**. For the measurement of Cytochorme C, blood samples were collected from the tail vain of RNU rats prior to the transplantation procedure and on days 3 and 7 after surgery. Cytochrome C was detected the serum samples using the Human Cytochrome C ELISA Kit (ab221832) following the manufacturer's instructions. Enolase 2 concentration was measured in whole organoid extracts using the human ENO2 ELISA kit (Proteintech, #KE00050), following the manufacturer's instructions.

**Enolase 2 activity**. Enolase 2 activity was measured in whole organoid extracts using the human ENO2 ELISA kit (Proteintech, #KE00050), following the manufacturer's instructions. During sample preparation, three organoids *per* sample were homogenized in 100 µL of ice-cold Enolase Assay Buffer. The extracts were centrifuged at 10,000 g for 5 min to remove insoluble material. Ten microliters of supernatant were transferred to the plate wells for measurement, to which 50 µL of the Reaction Mix were added to the wells. After incubating at 25 °C for 10 min, fluorescence intensity was read at $\lambda_{ex} = 570$ nm. Activity was calculated using the formula: Enolase Activity = (nmole of $H_2O_2$ × Sample Dilution Factor)/(Reaction Time) × Volume.

**Immunoprecipitation assay**. Immunoprecipitation experiments were performed using an amine-reactive resin-based method (Pierce, #26149), using antihuman p21$^{CIP1}$ (Novus Biologicals, #AF1047, 1:200). Immunoblotting was performed using the antibody antihuman pro-Caspase 3 from ThermoFisher Scientific (#31A893, 1:300). Primary antibodies were detected with peroxidase-conjugated antirabbit or anti-mouse IgG (1:3000) and visualized with SuperSignal West Femto Substrate (ThermoFisher Scientific #34094).

**Statistics**. Student's $t$ test, one-way ANOVA and two-way ANOVA were used. Where applicable, normal distribution of the data was determined using D'Agostino–Pearson and Shapiro–Wilk normality tests. Statistical analysis was performed using Prism 7 for Mac OS (GraphPad Software, Inc.).

**Reporting summary**. Further information on research design is available in the Nature Research Reporting Summary linked to this article.

## Data availability
The RNASeq datasets are available under GEO series record "GSE171474". Further information and data that support the findings of this study are available within the article and its Supplementary Information files or from the corresponding author upon reasonable request. A Source Data file with the raw results is provided with this paper. Source data are provided with this paper.

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

## Acknowledgements

This study was supported by grant funding from the National Institutes of Health (NIH) (R21AG058159 to DRL, R01DK124301 to DRL) and R01NS113591 to MS and the Intellectual and Developmental Disabilities Center at Boston Children's Hospital (BCH IDDRC; U54HD090255).

## Author contributions

J.O.R.H., X.W., M.V.S., M.L.M., M.F.S.R., A.M.H., D.O.L.C., and G.U.R.E. performed experimental studies. M.Su. generated iPSC lines. K.G., and C.K.P. performed data analysis. A.R., E.P.H. and D.J.K., M.Sa., and D.R.L. supervised different aspects of the work. D.R.L. wrote the article.

## Competing interests

The authors declare no competing interests.

## Additional information

**Peer review information** *Nature Communications* thanks Katalin Susztak and the other anonymous reviewer(s) for their contribution to the peer review this work. Peer reviewer reports are available.

