## [Peer Review File · Nature Communications]

Reviewers' Comments:

Reviewer #1:

Remarks to the Author:

The manuscript details the generation of kidney organoids from IPS cells from controls and from TSC mutations to model AML. This is an interesting application of the already established kidney organoid system developed more than 5 years ago. The authors claim that when implanted into mice, these organoids model AML better and they can be used for therapeutic testing. Overall the work is technically sound and the authors have generated multiple control and tsc null lines, on the other hand, I am sorry to say but I do think this is technically original. A lot of prior publications have used IPS cells engineered specific mutations and examined their phenotype and drug effects.

The maturity of these organoids remains unclear. The authors did not analyze the expression of mature podocytes or proximal tubule cells, only the expression of some basic transcription factors are shown, such as Lhx1, Hnf1, and Wt1. How does this compare to AML samples?

The expression of Acta2 is not specific for AML and expressed widely by stromal cells.

Figure3 is the most disturbing of all. There is only a 10% overlap of the DEG in the AML and normal kidneys and the TSC iPSC's. How can we claim that they are a good model for AML?

Figure4 cyst formation is remarkable. A variable definition is used for cysts, some call small dilated tubules cysts as well. PKD organoids form very few cysts, but here almost all organoids form a cyst. What was the definition of a cyst?

Figure 5 the degree of proliferation is remarkable. Such as 40% of cells are Ki67 positive. Is this something that is seen in patients with AML?

Overall I remain a bit skeptical about the resemblance of these organoids to AML. I think more sophisticated methods such as single-cell RNAseq would be essential to understand these cells.

Reviewer #2:

Remarks to the Author:

The authors demonstrate the feasibility of organoid models using transplanted hiPSC-derived to study TSC-associated kidney lesions and the possible development of novel therapies.

Comments:

1. In the introduction, the authors should delete the word "major" as obstacle to develop new therapy.
2. I suggest moving the last paragraph of introduction in the discussion.
3. I suggest rewriting the sentence of results, page 8 line 5. Melanocytic identity should be replaced by expression of melanogenesis markers. In general, I suggest replacing the word melanocytes markers with melanogenesis markers.
4. In the discussion, second paragraph, the authors should avoid the "mixed mesenchymal-melanocyte identity". The cell coexpresses smooth muscle and melanogenesis markers, and it is not proof of true differentiation identity. For instance, in the kidney, translocation renal cell carcinoma may express melanogenesis markers (HMB45 and Melan-A) and it is an epithelial neoplasm.
5. When the authors discussed AMLEC, they should also consider the so called "cambium layer" in which the expression of HMB45 and cathepsin K has been reported.
6. The authors should quote the manuscript of HMB45 in angiomyolipoma instead of epithelioid angiomyolipoma (ref 20, 21)

Reviewer #3:

Remarks to the Author:

The authors used hiPSC derived from patients with tuberous sclerosis complex to model the rare kidney tumor angiomyolipoma (AML) in the context of TSC2 mutations that lead to constitutive mTOR activation. Kidney organoids were generated from patient-derived hiPSC (TSC2+/-) as well as isogenic TSC2 -/-, and TSC2 +/+ lines. The transcriptional signature of TSC2-/- as assessed by bulk RNA-seq was comparable to that from AMLs. Furthermore, a myomelanocytic cell phenotype was observed and the development of epithelial cysts, as can be found in AML. Finally, organoids were transplanted in immunodeficient rats to test a new therapeutic approach using rapamycin-loaded nanoparticles.

Fig 1 > 2D part of the Morizane protocol showing MM and renal vesicle markers, nothing special, most of this could be supplemental in my opinion

Fig 2 > 3D part of the protocol, I miss some information on nephron segmentation (how much glom/prox/distal); especially in the -/- organoids I wonder whether these are still normal 'nephrons' containing more of these AML-cells or that nephron segmentation itself is affected.

- Interesting finding of cells expressing AML-markers PMEL and ACTA2 in -/- organoids ('AML-cells') which is absent in +/- or +/+.
- strong point > the expression is assessed by immunofluorescence as well as FACS and WB

Fig 3. > RNA profile is assessed by bulk RNA sequencing, showing the markers highly expressed in AML again in -/- organoids. Also, differential expression was confirmed to kidney AML tissue versus normal kidney.

- These data strengthen the disease model
- The authors do not use the disease model to answer new questions on biology. For example the organoids would form an ideal platform to isolate the AML-cells by means of FACS-sorting, using single cell RNA sequencing or making fluorescent reporters driven by one of the AML-specific markers.
- In this organoid model the AML cells probably arise from progenitors, which is more difficult to imagine for the disease as AML does not occur shortly after birth. Are ACTA2 or PMEL+ cells present at the MM or renal vesicle stage? Or is not derived from kidney progenitors at all?

Fig 4 > focused on cyst formation, which is increased in -/- organoids

- In the text, the authors refer to fig 4c about +/- which I think should be -/-
- The cysts in 4a seems to stain positive for LTL, PODXL and CADH1, which makes no sense in my opinion. The resolution of the images could be much better. The confocal images in 4e are more convincing and do not show PODXL.
- The legend of fig 4h is missing
- All in all, they characterize the cysts as being from 'tubular origin', which is not too original. I would argue the cysts found in the +/+ or +/- are probably from similar origin if they would stain those for LTL and ECAD. So they are just 'cysts' (but clearly present in higher frequency).

Fig.5 > transplantation of organoids/spheroids in rats; well characterized but not exciting, mTOR activation is assessed in cyst lining cells which looks convincing

- Also here I miss nephron segmentation data, I would be curious to see whether there are still normal glomeruli present or not.

Fig.6 > rapamycin treatment by subcapsular injection of rapa-loaded nanoparticles which results in reduced graft size, possibly due to apoptosis

- Why so complicated using subcapsular injection of rapa-loaded nanoparticles as people can get oral rapamycin or sirolimus for example? I personally do not see this as an improvement to current therapies.
 - Did rapamycin reduce cysts or also AML-cells outside the cysts?
 - There is hardly any data shown of the controls in fig 6 (b, d, e and f only show treated grafts)
- I can't find a reference to figure 7 in the text.

Minor issues

Er zitten hier en daar wat taalfoutjes en missers in het stuk (Angiomyolipoma in abstract; er mist een) bij (ACTA2 ; p12 > the spatial the analysis of spatial....)

Reviewer #1 (Remarks to the Author):

Reviewer: “The manuscript details the generation of kidney organoids from IPS cells from controls and from TSC mutations to model AML. This is an interesting application of the already established kidney organoid system developed more than 5 years ago. The authors claim that when implanted into mice, these organoids model AML better and they can be used for therapeutic testing.

Overall the work is technically sound and the authors have generated multiple control and tsc null lines, on the other hand, I am sorry to say but I do think this is technically original. A lot of prior publications have used IPS cells engineered specific mutations and examined their phenotype and drug effects.”

Answer: We thank the reviewer for the time taken to help us improve our manuscript. We are grateful for the positive comments including “the work is technically sound and the authors have generated multiple control and tsc null lines”.

In terms of originality, we would like to highlight four key technical and translational original advances of this work:

1- This is the first study to model human renal disease *in vivo*, using iPSC-derived organoids. We have created a preclinical animal model that preserves the biological properties and phenotypes observed *in vitro*, including the key cellular and molecular aspects of AML cells, as well as nephron tubule cysts. This achievement represents a key landmark for the utilization of kidney organoids in biomedical research.

2- We have recapitulated the renal manifestations of a rare genetic disease, TSC, using iPSCs. Efforts to reliably reproduce angiomyolipomata in animal models have failed for more than 20 years. This has been a major obstacle to the understanding of tumor mechanisms and has hampered the development of novel therapies.

3- We have used an uncommon orthogonal approach involving *a battery of techniques*, used *in vitro* and *in vivo*, to validate the phenotype of our organoids, increasing the robustness of our conclusions.

4- Our results experimentally demonstrate for the first time a causal relationship between early developmental loss of *TSC2* and the spectrum of kidney tissues and cellular lineages that affected. This is likely to explain the combination of angiomyolipomata and cystic kidney disease that is observed in the kidneys of many TSC patients. This finding is of high importance and interest both scientifically and medically.

In order to address the Reviewer's concern regarding the similarities between *TSC2*^{-/-} organoids and kidney AML, we have included new experiments that show: **1)** analysis of AML gene expression in the purified iPSC-derived myoid AML cell population in new panel Figure 2f, validating the myomelanocytic AML cell identity of iPSC-derived GPNMB-expressing cells; **2)** analysis of AML transcription factors in new panel Figure 1g, **3)** evidence of overexpression and increased activity of the glycolytic enzyme enolase 2, which is a *bona fide* neuroendocrine tumor biomarker upregulated in AML tumors of patients; and **4)** evidence for a molecular mechanism of tumor resistance in AML, that was investigated in *TSC2*^{-/-} AML organoid xenografts (new Figure 7). We hope that, with the addition of these new results, the body of data now provides sufficient evidence of the similarities between *TSC2*^{-/-} iPSC-derived renal organoids and TSC-associated renal AML tumors.

Reviewer: The maturity of these organoids remains unclear. The authors did not analyze the expression of mature podocytes or proximal tubule cells, only the expression of some basic transcription factors are shown, such as Lhx1, Hnf1, and Wt1. How does this compare to AML samples?

Answer: We have now included a new panel (Supplemental figure 5a showing RNAseq data for several major mature nephron segment markers, and included a paragraph in the main text stating "These phenotypic characteristics were not associated with changes in the expression of nephron segment markers such as *PODXL*, *NPHS1* and *SYNPO* (glomerulus); *AQP1* and *CDH2* (proximal tubule); or *EPCAM* and *CDH1* (distal tubule), whose mRNAs were detected at comparable levels in our RNAseq analysis of *TSC2*^{+/+}, *TSC2*^{+/-} and *TSC2*^{-/-} organoids (Supplementary figure 5a)".

We want to highlight that the presence of podocytes and glomeruli, proximal tubules, and distal segments nephron segments is evident in Figure 4, where podocyte-like cells are stained with PDXL1, proximal tubules labeled with lotus tetranoglobus lectin (LTL), and distal tubules expressing CDH1 are shown with various degree of detail in *TSC2*^{-/-} organoids (panels b, c, e, f and g). Particularly, Panel e in Figure 4 shows the three components together in *TSC2*^{-/-} organoids.

Comparing nephron segment content between *TSC2*^{-/-} AML organoids and kidney AML is not possible, as pure patient-derived AML samples do not contain mature podocytes or proximal tubule cells, as shown in previous studies by IHC.

Reviewer: "The expression of Acta2 is not specific for AML and expressed widely by stromal cells."

Answer: We do not intend to affirm that *ACTA2* is exclusively expressed in AML cells, but that in *TSC2*^{-/-} organoids and in AML tumors *ACTA2* expression is aberrant compared to normal kidney or to control organoids, making *ACTA2* a widely-used

diagnostic marker of AML. Our data shows that *ACTA2* expression is significantly higher in *TSC2*^{-/-} organoids and in AML tumors compared to normal kidney and control organoids, and that it occurs in a population of cells that additionally co-express melanocyte genes. We show this with multiple techniques in Figure 3 b, 3d, and 3e, as well as in the new panel 3f. In contrast, expression of melanocyte genes does not occur in normal kidney stromal cells, nor does it occur the *ACTA2*⁺ cells of *TSC2*^{+/+} and *TSC2*^{+/-} organoids as shown in Figure 3b, 3d and 3e.

Lastly, in normal uninjured kidneys and in uninjured renal organoids, *ACTA2* is expressed at low levels by stromal pericytes and fibroblasts, only becoming upregulated in response to injury or in the context of chronic disease, as part of the process of differentiation to myofibroblast (see Kramann R *et al.* Cell Stem Cell. 2015 Jan 8;16(1):51-66 and Lemos DR *et al.* J Am Soc Nephrol. 2018 Jun;29(6):1690-1705). Accordingly, in Figure 2d, we show that even in the injured renal organoid, *ACTA2*-expressing myofibroblasts do not express the melanocyte protein GPNMB.

Reviewer: Figure3 is the most disturbing of all. There is only a 10% overlap of the DEG in the AML and normal kidneys and the TSC IPSC's. How can we claim that they are a good model for AML?

Answer: To clarify, the result is in fact consistent with the fact that among the genes found to be differentially expressed between normal kidney and kidney AML (i.e. *in vivo*), there are several genes involved in immune function, which the organoids lack. Additionally, there are genes regulated by systemic signals that are also not present in the organoid cultures.

The value of Figure 3e, therefore, resides in the unbiased identification of AML signature genes differentially expressed in both comparisons. The finding, together with the analysis of signaling pathways in Figure 3d, and the comparison of FPKMs shown in panel f, is key evidence supporting the AML phenotype of *TSC2*^{-/-} organoids on a molecular level.

Reviewer: Figure4 cyst formation is remarkable. A variable definition is used for cysts, some call small dilated tubules cysts as well. PKD organoids form very few cysts, but here almost all organoids form a cyst. What was the definition of a cyst?

Answer: We generally do not observe multiple cysts within individual *TSC2*^{-/-} organoids, rather we commonly find single cysts, examples of this are shown in Figure 4 panel b, panel d and panel e. Anatomical and molecular analysis of the cysts shows three characteristics associated with kidney cysts described in TSC patients: **1)** Loss of cell polarity (shown and quantified in Figure 4g); **2)** proliferative activity a shown in Figure 5j and 5l, which is consistent with the proliferative activity described for PKD by multiple authors; **3)** alternating stretches of columnar epithelium as shown in Figure 4f, that closely resembles the stretches of columnar

epithelium also found in cysts of TSC patients, well described by Stapleton FB, *et al.* (1980), *The Journal of pediatrics* 97(4):574-579.

Reviewer: Figure 5 the degree of proliferation is remarkable. Such as 40% of cells are Ki67 positive. Is this something that is seen in patients with AML?

Answer: To clarify, in current Figure 5l (previously panel 5k), the graph shows 40% of proliferation for cystic tubule epithelial cells, not for AML cells. The percentage of proliferation for AML cells is less than 10%.

Reviewer: Overall I remain a bit skeptical about the resemblance of these organoids to AML. I think more sophisticated methods such as single-cell RNAseq would be essential to understand these cells.

Answer: We agree that future studies, potentially including comparative scRNA seq analyses of *TSC2*^{-/-} iPSC-derived AML cells *versus* kidney AML cells, may help to further validate our organoid model.

In this manuscript we provide a technically sound body of evidence showing that *TSC2*^{-/-} renal organoids contain myomelanocytic cells with characteristic of AML cells. We used an orthogonal approach that involves multiple techniques, including **1)** flow cytometry, **2)** immunohistochemistry, and **3)** expression and activity of a glycolytic enzyme upregulated in AML, that is a *bona fide* tumor biomarker (new Figure 3g, h, i), **4)** newly added experiments of AML cell purification followed by gene expression analysis of AML genes (new Figure 2f). In particular AML cell purification followed by PCR analysis is more robust than single cell RNAseq analysis, and better suited to our current research budget; and **5)** mechanistic evidence supporting the similarities between *TSC2*^{-/-} organoids and AML tumors on a molecular level, through the identification of a prospective mechanism of tumor resistance involving the gene *CDKN1A* in kidney AML and in *TSC2*^{-/-} organoids (shown in new Figure 7). This finding further highlights the biological relevance of *TSC2*^{-/-} organoids as an experimental model of AML with potential therapeutic relevance.

Reviewer #2 (Remarks to the Author):

The authors demonstrate the feasibility of organoid models using transplanted hiPSC-derived to study TSC-associated kidney lesions and the possible development of novel therapies.

Answer: We sincerely thank the reviewer for the constructive comments.

Comments:

Reviewer: 1. In the introduction, the authors should delete the word “major” as obstacle to develop new therapy.

Answer: Thank you, we have deleted the word.

Reviewer: 2. I suggest moving the last paragraph of introduction in the discussion.

Answer: Thank you for this suggestion. After careful consideration of the format for the last paragraph of the Introduction as suggested by the Journal, and after examination of similar articles, we decided to leave the paragraph as a bridge between the Introduction and the Results section. We would be happy to re-address based on the Editor’s opinion.

Reviewer: 3. I suggest rewriting the sentence of results, page 8 line 5. Melanocytic identity should be replaced by expression of melanogenesis markers. In general, I suggest replacing the word melanocytes markers with melanogenesis markers.

Answer: Thank you. We have changed “melanocytic identity” to “melanogenic phenotype.” in the mentioned line.

Reviewer: 4. In the discussion, second paragraph, the authors should avoid the “mixed mesenchymal-melanocyte identity”. The cell coexpresses smooth muscle and melanogenesis markers, and it is not proof of true differentiation identity. For instance, in the kidney, translocation renal cell carcinoma may express melanogenesis markers (HMB45 and Melan-A) and it is an epithelial neoplasm.

Answer: Thank you. We have changed the term to “dual smooth muscle and melanocyte cell phenotype”. We agree that melanogenesis markers may not necessarily indicate developmental adoption of a melanocyte fate *during differentiation*. Indeed the RNAseq data shown in new Figure 1g shows upregulation of the AML driver oncogene MITF, and generation of morphologically myoid PMEL-expressing cells throughout the protocol of nephric differentiation (Figure 1h).

Reviewer: 5. When the authors discussed AMLEC, they should also consider the so called “cambium layer” in which the expression of HMB45 and cathepsin K has been reported.

Answer: We now mention the sub-epithelial cambium AML cells at the point in the discussion where we refer to AMLEC (see manuscript text highlighted in the discussion).

Reviewer: 6. The authors should quote the manuscript of HMB45 in angiomyolipoma instead of epithelioid angiomyolipoma (ref 20, 21)

Answer: Thank you. We have modified those references as the reviewer suggested.
Thanks

Reviewer #3 (Remarks to the Author):

The authors used hiPSC derived from patients with tuberous sclerosis complex to model the rare kidney tumor angiomyolipoma (AML) in the context of TSC2 mutations that lead to constitutive mTOR activation. Kidney organoids were generated from patient-derived hiPSC (TSC2+/-) as well as isogenic TSC2 -/- , and TSC2 +/+ lines. The transcriptional signature of TSC2-/- as assessed by bulk RNA-seq was comparable to that from AMLs. Furthermore, a myomelanocytic cell phenotype was observed and the development of epithelial cysts, as can be found in AML. Finally, organoids were transplanted in immunodeficient rats to test a new therapeutic approach using rapamycin-loaded nanoparticles.

Reviewer: Fig 1 > 2D part of the Morizane protocol showing MM and renal vesicle markers, nothing special, most of this could be supplemental in my opinion

Answer: We have now modified Figure 1. We removed the SIX2 immunofluorescence panel (now Supplementary Figure 1a) and have added upregulation of the AML oncogenic driver MITF in panel g, as well as generation of PMEL-expressing myoid cells on Days 14 and 21 of differentiation in panel h. These results are relevant for the subsequent AML organoid data shown in Figure 2.

Reviewer: Fig 2 > 3D part of the protocol, I miss some information on nephron segmentation (how much glom/prox/distal); especially in the -/- organoids I wonder whether these are still normal 'nephrons' containing more of these AML-cells or that nephron segmentation itself is affected.

Answer: We are happy to clarify. In the section titled "*TSC2 inactivation drives renal organoid tubule cyst formation*", we show that the three segments are clearly segregated in the 3-D *TSC2*^{-/-} 3-D organoid (Figure 4e). Overall, we observe segmentation, albeit the analysis is made complicated by the cysts. As mentioned in the second paragraph of the same section of Results, in the case of the cysts, the lining mainly comprises cells from individual tubule segments, indicating

segregation, with short stretches where the cysts are lined by a combination of both proximal and distal tubule cells. The latter might indicate cystogenesis occurring in areas of transition from proximal to distal segments, with cystogenesis possibly affecting the segregation.

Lastly, and, as also mentioned at the end of the first paragraph of the abovementioned Results section, glomeruli are present in *TSC2*^{-/-} organoids (see Figure 4b, c and e), albeit the arrangement of PDXL1⁺ podocyte cells *in vitro* is less compact than in the glomeruli of *TSC2*^{+/+} and *TSC2*^{+/-} organoids, suggesting that glomerular anatomy is somewhat affected by the loss of *TSC2*, this is something that we are currently studying.

The overall conclusion is that the nephrons in the *TSC2*^{-/-} organoids contain all three main segments described, and these can be clearly seen well segregated in 3-D cultures.

Reviewer: - Interesting finding of cells expressing AML-markers PMEL and ACTA2 in -/- organoids ('AML-cells') which is absent in +/- or +/+.

Answer: Thank you. We agree that this is a remarkable finding.

Reviewer: - strong point > the expression is assessed by immunofluorescence as well as FACS and WB

Answer: Thank you. We agree that this is an important result. In addition the manuscript now includes an additional experiment in which we purified the organoid AML cell population and validated AML gene expression using RT-PCR. These results are reported in new panel f of Figure 2.

Reviewer: Fig 3. > RNA profile is assessed by bulk RNA sequencing, showing the markers highly expressed in AML again in -/- organoids. Also, differential expression was confirmed to kidney AML tissue versus normal kidney.

- These data strengthen the disease model

- The authors do not use the disease model to answer new questions on biology. For example the organoids would form an ideal platform to isolate the AML-cells by means of FACS-sorting, using single cell RNA sequencing or making fluorescent reporters driven by one of the AML-specific markers.

Answer: Thank you. We agree that our RNAseq data strengthen the AML organoid model. Regarding the use of this model to answer AML biology questions, we have now added a new Figure 7 in which we have used our *TSC2*^{-/-} organoids to identify a mechanism of tumor resistance driven by overexpression of *CDKN1A* and stabilization of its protein p21^{CIP1} in the cytoplasm AML cells of TSC patients. Our *in vivo* experiments using organoid xenografts indicated that p21^{CIP1} directly interacts

with pro-caspase 3 and this interaction can be disrupted by rapamycin-nanoparticles, leading to apoptosis activation in AML cells, adding mechanistic insight to the results of Figure 6. This finding is of biological relevance and offers cues toward the clinical problem of why rapalog treatments generally fail to ablate AML. Based on these findings, we are currently designing novel nanoparticle-based therapies that can override tumor resistance mechanisms in a tissue-specific manner.

Indeed, we are planning single cell RNAseq experiments that will serve to compare gene expression in AML-like cells from iPSC-derived organoids versus AML cells from tumor samples. These experiments will likely be part of a future study.

Reviewer: - In this organoid model the AML cells probably arise from progenitors, which is more difficult to imagine for the disease as AML does not occur shortly after birth. Are ACTA2 or PMEL⁺ cells present at the MM or renal vesicle stage? Or is not derived from kidney progenitors at all?

Answer: We agree that this is an important point. The origin of AML cells in the kidney remains a topic of discussion in the field. We are currently investigating the developmental origin of PMEL⁺/GPNMB⁺ ACTA2⁺ cells in our iPSC-derived cell cultures, and are in the process of creating new reporter iPSC lines to genetically label AML cells and trace their developmental origin. In this revised version of the article, we have added a new data shown in Figure 1 panel g reporting upregulation of the AML master driver transcription factor MITF, and generation of myoid PMEL-expressing cells during the protocol of nephric differentiation, firstly detected around Day 14, at the time of renal vesicle formation (new panel h of Figure 1). In addition, in Supplementary Figure 2 panel we show the detection of GPNMB expression in ACTA2⁺ myomelanocytic cells in our 2D differentiating cultures. Overall, our results support the idea of an aberrant course of differentiation in the absence of *TSC2*. We are actively investigating the underlying mechanisms.

Another important issue is that we now understand that renal AMLs take much longer to develop than what was previously thought. AML are seen in patients as young as 10 years old, and it is very possible that these lesions begin during early renal development, and their growth is enhanced during puberty.

Reviewer: Fig 4 > focused on cyst formation, which is increased in -/- organoids - In the text, the authors refer to fig 4c about +/- which I think should be -/-

Answer: Many thanks for pointing out this mistake. We have corrected the text to correctly indicate *TSC2*^{-/-}.

Reviewer:- The cysts in 4a seems to stain positive for LTL, PODXL and CADH1, which makes no sense in my opinion. The resolution of the images could be much better. The confocal images in 4e are more convincing and do not show PODXL.

Answer: Thank you for pointing this out this error. The green channel was showing Vimentin, not PODXL1. We have now placed the correct staining showing a cluster of PODXL1+ cells, associated with the cyst. The resolution of the image should improve in the final version of the manuscript, the previous version was a reduced size pdf file. Also, in this version, the cluster of green PODXL1+ cells should be visible in panel 4e (*TSC2*^{-/-} 3D organoid section).

Reviewer:- The legend of fig 4h is missing

Answer: Thanks very much for bringing this to our attention. We have added the legend for Figure 4h.

Reviewer:- All in all, they characterize the cysts as being from ‘tubular origin’, which is not too original. I would argue the cysts found in the +/+ or +/- are probably from similar origin if they would stain those for LTL and ECAD. So they are just ‘cysts’ (but clearly present in higher frequency).

Answer: The cystic phenotype observed in TSC patients is largely associated with the nephron tubules. Our organoids reproduce that cystic phenotype. We also show that cystogenesis can occur in the presence of *PKD1* expression, suggesting that the mechanisms are different from those driving PKD.

To clarify, we very rarely observe cystic structures in either *TSC2*^{+/+} or *TSC2*^{+/-} organoids. Cystic disease in TSC is phenotypically different than in PKD, and therefore, this is the first model of cystogenesis in TSC, using organoids. But perhaps what is most relevant about the co-occurrence of AML and cystic disease simultaneously in our organoids is that it demonstrates experimentally *for the first time* the causal relationship between early developmental loss of *TSC2* and spectrum of kidney tissues affected, explaining the combination of angiomyolipomata and cystic kidney disease that is observed in the kidneys of many TSC patients.

Another key and original difference between our model and other models of cystic disease is that the cysts in the TSC model form spontaneously, in contrast to other cystic kidney models in which cAMP analogues are used.

Reviewer: Fig.5 > transplantation of organoids/spheroids in rats; well characterized but not exciting, mTOR activation is assessed in cyst lining cells which looks convincing

Answer: In terms of whether these transplantation data of the organoids into rats are exciting, we would like to highlight that our model recapitulates for the first time a human renal disease using iPSC-derived organoids. This is especially impactful for the TSC field, which has never had an *in vivo* model of angiomyolipomas, despite at least two decades of effort. We believe that the data represent a landmark in terms of disease modeling with iPSCs.

Figure 4g shows mTOR activation in PMEL⁺ AML-like cells as well.

Reviewer:- Also here I miss nephron segmentation data, I would be curious to see whether there are still normal glomeruli present or not.

Answer: We certainly observe PODXL1⁺ cells organized into glomerulus-like structures in transplanted *TSC2*^{-/-} organoids. We have now added a representative example in Figure 5 panel f.

Reviewer: Fig.6 > rapamycin treatment by subcapsular injection of rapa-loaded nanoparticles which results in reduced graft size, possibly due to apoptosis - Why so complicated using subcapsular injection of rapa-loaded nanoparticles as people can get oral rapamycin or sirolimus for example? I personally do not see this as an improvement to current therapies.

Answer: Our results are proof-of-concept that rapalog nanotherapy may be beneficial in selectively targeting AML with minimal effect on other tissues. In addition, we show that the *in vivo* AML organoid model can be used to test new therapeutic applications, which is something that could not be done until now for this disease.

Reviewer: - Did rapamycin reduce cysts or also AML-cells outside the cysts?

Answer: The rapa-nanoparticles reduced cyst size dramatically, but we were not able to quantify the size of cyst remnants, which contained scattered tubule cells with pyknotic nuclei. This is shown in new Supplementary Figure 7a. We mention this finding in the Results section "Delivery of rapamycin-loaded nanoparticles abrogates orthotopic *TSC2*^{-/-} AML xenografts."

Reviewer:- There is hardly any data shown of the controls in fig 6 (b, d, e and f only show treated grafts)

Answer: Thank you for bringing this to our attention, Figure 6 panel f shows quantification of Cyt C in both treated and non-treated xenograft and we have now added new Supplementary Figure 7 panels b and c, showing no activation of Caspase 3 or DNA fragmentation in non-treated samples at Day 7 of treatment. The quantifications have been incorporated into the new bar graphs displayed in Figure 6 panels d and h.

Reviewer: I can't find a reference to figure 7 in the text.

Answer: The previous version of Figure 7 has now been replaced for a new Figure 7 identifying a molecular mechanism in *TSC2*^{-/-} organoids, with its corresponding legend. The previous Figure 7 depicting a schematic of our disease modeling approach, has been removed.

Reviewer: Minor issues

Er zitten hier en daar wat taalfoutjes en missers in het stuk (Angiomyolipoma in abstract; er mist een) bij (ACTA2 ; p12 > the spatial the analysis of spatial....)

Answer: Thanks very much for pointing out these errors. We have made corrections accordingly

In closing, we again thank the four Reviewers and the Editor for their many positive comments and for their suggestions, all of which we have worked hard to address.

Reviewers' Comments:

Reviewer #1:

Remarks to the Author:

My concerns have been answered

Reviewer #2:

Remarks to the Author:

The authors have satisfactorily responded to all my comments.

Reviewer #3:

Remarks to the Author:

The paper is acceptable for publication now